# Dynamic Diffusion Transformer

**Wangbo Zhao**[1*] **Yizeng Han**[2] **Jiasheng Tang**[2,3†] **Kai Wang**[1] **Yibing Song**[2,3]

**Gao Huang**[4] **Fan Wang**[2] **Yang You**[1†]

[1]National University of Singapore  [2]DAMO Academy, Alibaba Group
[3]Hupan Lab  [4]Tsinghua University

## Abstract

Diffusion Transformer (DiT), an emerging diffusion model for image generation, has demonstrated superior performance but suffers from substantial computational costs. Our investigations reveal that these costs stem from the *static* inference paradigm, which inevitably introduces redundant computation in certain *diffusion timesteps* and *spatial regions*. To address this inefficiency, we propose **Dy**namic **Di**ffusion **T**ransformer (DyDiT), an architecture that *dynamically* adjusts its computation along both *timestep* and *spatial* dimensions during generation. Specifically, we introduce a *Timestep-wise Dynamic Width* (TDW) approach that adapts model width conditioned on the generation timesteps. In addition, we design a *Spatial-wise Dynamic Token* (SDT) strategy to avoid redundant computation at unnecessary spatial locations. Extensive experiments on various datasets and different-sized models verify the superiority of DyDiT. Notably, with <3% additional fine-tuning iterations, our method reduces the FLOPs of DiT-XL by 51%, accelerates generation by $1.73\times$, and achieves a competitive FID score of 2.07 on ImageNet. The code is publicly available at **https://github.com/NUS-HPC-AI-Lab/Dynamic-Diffusion-Transformer**.

## 1 Introduction

Diffusion models (Ho et al., 2020; Dhariwal & Nichol, 2021; Rombach et al., 2022; Blattmann et al., 2023; He et al., 2025) have demonstrated significant superiority in visual generation. Recently, the remarkable scalability of Transformers (Vaswani et al., 2017; Dosovitskiy et al., 2020) has led to the growing prominence of Diffusion Transformer (DiT) (Peebles & Xie, 2023). DiT has shown strong potential across various tasks (Chen et al., 2023; Ma et al., 2024b; Chen et al., 2024; Nan et al., 2024) and is considered as a foundational component of Sora (Brooks et al., 2024), a pioneering model for video generation. Like Transformers in other domains (Dosovitskiy et al., 2020; Brown et al., 2020; Ni et al., 2024b;a; Han et al., 2024b;a), DiT faces significant efficiency challenges during generation.

Existing approaches to improving DiT's efficiency include efficient diffusion samplers (Song et al., 2020a; 2023; Salimans & Ho, 2022; Meng et al., 2023; Luo et al., 2023) and global acceleration techniques (Ma et al., 2023; Pan et al., 2024). In addition, reducing computational redundancy within the DiT architecture using model compression techniques, such as structural pruning (Fang et al., 2024; Molchanov et al., 2016; He et al., 2017), also shows significant promise.

However, pruning methods typically retain a *static* architecture across both the *timestep* and *spatial* dimensions throughout the diffusion process. As shown in Figure 1(c), both the original DiT and the pruned DiT employ a fixed model width across all diffusion timesteps and allocate the same computational cost to every image patch. This static inference paradigm overlooks the varying complexities associated with different timesteps and spatial regions, leading to significant computational inefficiency. To explore this redundancy in more detail, we analyze the training process of DiT, during which it is optimized for a noise prediction task. Our analysis yields two key insights:

*a) Timestep perspective:* We plot the loss value differences between a pre-trained small model (DiT-S) and a larger model (DiT-XL) in Figure 1(a). The results show that the loss differences diminish substantially for $t > \hat{t}$, and even approach negligible levels as $t$ nears the prior distribution ($t \rightarrow T$).

---

[1]Work done during an internship at DAMO Academy, Alibaba Group, wangbo.zhao96@gmail.com

[2]Corresponding authors, jiasheng.tjs@alibaba-inc.com, youy@comp.nus.edu.sg

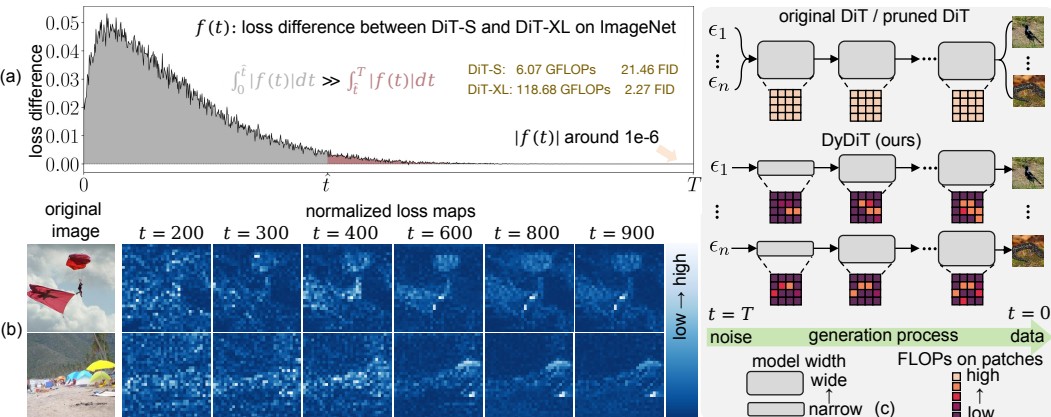

Figure 1: (a) The loss difference between DiT-S and DiT-XL across all diffusion timesteps ($T = 1000$). The difference is slight at most timesteps. (b) Loss maps (normalized to the range [0, 1]) at different timesteps, show that the noise in different patches has varying levels of difficulty to predict. (c) Difference of the inference paradigm between the static DiT and the proposed DyDiT.

This indicates that the prediction task becomes *progressively easier at later timesteps* and could be managed effectively even by *a smaller model*. However, DiT applies the same architecture across all timesteps, leading to *excessive computational costs at timesteps where the task complexity is low*.

*b) Spatial perspective:* We visualize the loss maps in Figure 1(b) and observe a noticeable imbalance in loss values across different spatial regions of the image. Loss values are higher in patches corresponding to the main object, while patches representing background regions exhibit relatively lower loss. This suggests that the difficulty of noise prediction varies across spatial regions. Consequently, *uniform computational treatment of all patches introduces redundancy and is likely suboptimal*.

Based on the above insights, a promising approach to improve DiT's computational efficiency is *dynamic computation*. To this end, we propose **Dy**namic **Di**ffusion **T**ransformer (DyDiT), which adaptively allocates computational resources during the generation process, as illustrated in Figure 1(c). Specifically, from the timestep perspective, we introduce a *Timestep-wise Dynamic Width* (TDW) mechanism, where the model learns to adjust the width of the attention and MLP blocks based on the current *timestep*. From a spatial perspective, we develop a *Spatial-wise Dynamic Token* (SDT) strategy, which identifies image patches where noise prediction is relatively "easy", allowing them to bypass computationally intensive blocks, thus reducing unnecessary computation.

Notably, both TWD and SDT are plug-and-play modules that can be easily implemented on DiT to build DyDiT. Moreover, our method contributes to significant speedup due to the hardware-friendly design: 1) the model architecture at each timestep can be pre-determined offline, eliminating additional overhead for width adjustments and enabling efficient batch processing (Section 3.2); and 2) the token gathering and scattering operations incur minimal overhead and are straightforward to implement (Section 3.3). Such hardware efficiency distinguishes our approach from traditional dynamic networks (Herrmann et al., 2020; Meng et al., 2022; Han et al., 2024c), which adapt their inference graphs for each sample and struggle to improve practical efficiency in batched inference.

We conduct extensive experiments across multiple datasets and model scales to validate the effectiveness of the proposed method. For example, compared to the static counterpart DiT-XL, our DyDiT-XL reduces FLOPs by 51% and accelerates the generation by 1.73 times, with less than 3% fine-tuning iterations, while maintaining a competitive FID score of 2.07 on ImageNet ($256 \times 256$) (Deng et al., 2009). Our method shows potential for further efficiency gains when combined with efficient samplers, such as DDIM (Song et al., 2020a) and DPM Solver++ (Lu et al., 2022), or global acceleration techniques like DeepCache (Ma et al., 2023). We anticipate that DyDiT will inspire future research in the development of more efficient diffusion Transformers.

## 2 RELATED WORKS

**Efficient Diffusion Models.** Although diffusion models (Ho et al., 2020; Rombach et al., 2022) have achieved remarkable performance in generation tasks, their generation speed has always hindered

their further applications primarily due to long sampling steps and high computational costs. Existing attempts to make diffusion models efficient can be roughly categorized into sampler-based methods, model-based methods, and global acceleration methods. The sampler-based methods (Song et al., 2020a; 2023; Salimans & Ho, 2022; Meng et al., 2023; Luo et al., 2023) aim to reduce the sampling steps. Model-based approaches (Fang et al., 2024; So et al., 2024; Shang et al., 2023; Yang et al., 2023; Pu et al., 2024) attempt to compress the size of diffusion models via pruning (Fang et al., 2024; Shang et al., 2023) or quantization (Li et al., 2023; Shang et al., 2023). Global acceleration methods like Deepcache (Ma et al., 2023) tend to reuse or share some features across different timesteps.

Our DyDiT is mostly relates to the model-based approaches and orthoganal to the other two lines of work. However, unlike the pruning methods yielding *static architectures*, DyDiT performs *dynamic computation* for different diffusion timesteps and image tokens.

**Dynamic Neural Networks.** Compared to static models, dynamic neural networks (Han et al., 2021) can adapt their computational graph based on inputs, enabling superior trade-off between performance and efficiency. They generally realize dynamic architectures by varying the network depth (Teerapittayanon et al., 2016; Bolukbasi et al., 2017; Yang et al., 2020; Han et al., 2022; 2023) or width (Herrmann et al., 2020; Li et al., 2021; Han et al., 2024c) during inference. Some works explore the spatial redundancy in image *recognition* (Wang et al., 2020; 2021; Song et al., 2021; Rao et al., 2021; Liang et al., 2022; Meng et al., 2022). Despite their promising theoretical efficiency, existing dynamic networks usually struggle in achieving *practical efficiency* during batched inference (Han et al., 2024c) due to the per-sample inference graph. Moreover, the potential of dynamic architectures in diffusion models, where a *timestep* dimension is introduced, remains unexplored.

This work extends the research of dynamic networks to the image *generation* field. More importantly, our TDW adjusts the network structure only conditioned on the *timesteps*, avoiding the sample-conditioned tensor shapes in batched inference. Together with the efficient token gathering and scattering mechanism of SDT, DyDiT shows preferable realistic efficiency.

## 3 DYNAMIC DIFFUSION TRANSFORMER

We first provide an overview of diffusion models and DiT (Peebles & Xie, 2023) in Section 3.1. DyDiT's timestep-wise dynamic width (TDW) and spatial-wise dynamic token (SDT) approaches are then introduced in Sections 3.2 and 3.3. Finally, Section 3.4 details the training process of DyDiT.

### 3.1 PRELIMINARY

**Diffusion Models** (Ho et al., 2020; Song et al., 2020b; Nichol & Dhariwal, 2021; Rombach et al., 2022) generate images from random noise through a series of diffusion steps. These models typically consist of a forward diffusion process and a reverse denoising process. In the forward process, given an image $\mathbf{x}_0 \sim q(\mathbf{x})$ sampled from the data distribution, Gaussian noise $\epsilon \sim \mathcal{N}(0, I)$ is progressively added over $T$ steps. This process is defined as $q\left(\mathbf{x}_t \mid \mathbf{x}_{t-1}\right) = \mathcal{N}\left(\mathbf{x}_t; \sqrt{1 - \beta_t}\mathbf{x}_{t-1}, \beta_t\mathbf{I}\right)$, where $t$ and $\beta_t$ denote the timestep and noise schedule, respectively. In the reverse process, the model removes the noise and reconstructs $\mathbf{x}_0$ from $\mathbf{x}_T \sim \mathcal{N}(0, I)$ using $p_\theta\left(\mathbf{x}_{t-1} \mid \mathbf{x}_t\right) = \mathcal{N}\left(\mathbf{x}_{t-1}; \mu_\theta\left(\mathbf{x}_t, t\right), \Sigma_\theta\left(\mathbf{x}_t, t\right)\right)$, where $\mu_\theta\left(\mathbf{x}_t, t\right)$ and $\Sigma_\theta(\mathbf{x}_t, t)$ represent the mean and variance of the Gaussian distribution.

**Diffusion Transformer** (DiT) (Peebles & Xie, 2023) exhibits the scalability and promising performance of Transformers (Brooks et al., 2024), as theoretically supported by Hu et al. (2024a;b). Similar to ViT (Dosovitskiy et al., 2020), DiT consists of layers composed of a multi-head self-attention (MHSA) block and a multi-layer perceptron (MLP) block, described as $\mathbf{X} \leftarrow \mathbf{X} + \alpha\text{MHSA}(\gamma\mathbf{X} + \beta), \mathbf{X} \leftarrow \mathbf{X} + \alpha'\text{MLP}(\gamma'\mathbf{X} + \beta')$, where $\mathbf{X} \in \mathbb{R}^{N \times C}$ denotes image tokens. Here, $N$ is the number of tokens, and $C$ is the channel dimension. The parameters $\{\alpha, \gamma, \beta, \alpha', \gamma', \beta'\}$ are produced by an adaptive layer norm (adaLN) block (Perez et al., 2018), which takes the class condition embedding $\mathbf{E}_{cls}$ and timestep embedding $\mathbf{E}_t$ as inputs.

### 3.2 TIMESTEP-WISE DYNAMIC WIDTH

As aforementioned, DiT spends equal computation for different timesteps, although not all steps share the same generation difficulty (Figure 1(a)). Therefore, the *static* computation paradigm introduces significant redundancy in those "easy" timesteps. Inspired by structural pruning methods (He et al., 2017; Hou et al., 2020; Fang et al., 2024), we propose a *timestep-wise dynamic width* (**TDW**) mechanism, which adjusts the width of MHSA and MLP blocks in different timesteps. Note that

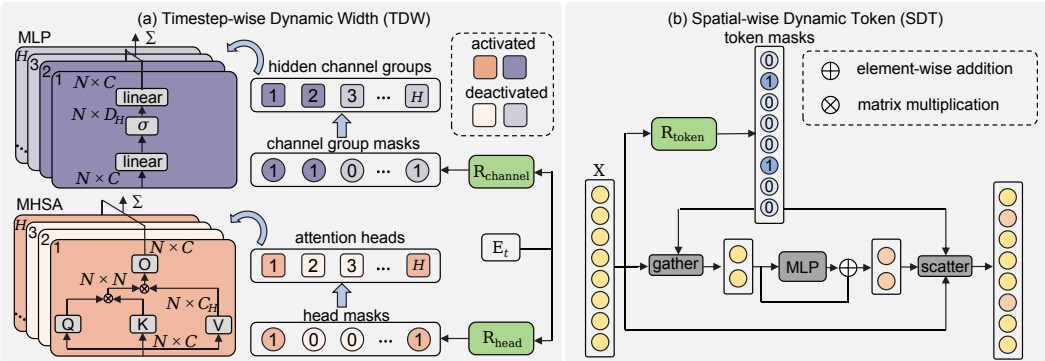

Figure 2: **Overview of the proposed dynamic diffusion transformer (DyDiT)**. It reduces the computational redundancy in DiT (Peebles & Xie, 2023) from both timestep and spatial dimensions.

TDW is *not a pruning method* that permanently removes certain model components, but rather retains the full capacity of DiT and *dynamically* activates different heads/channel groups at each timestep.

**Heads and channel groups.** Given input $\mathbf{X} \in \mathbb{R}^{N \times C}$, an MHSA block employs three linear layers with weights $\mathbf{W}_Q, \mathbf{W}_K, \mathbf{W}_V \in \mathbb{R}^{C \times (H \times C_H)}$ to project it into Q, K, and V features, respectively. Here, $H$ denotes the head number and $C = H \times C_H$ in DiT. An output projection is performed using another linear layer with $\mathbf{W}_O \in \mathbb{R}^{(H \times C_H) \times C}$. The operation of the conventional MHSA can be expressed as:

$$\text{MHSA}(\mathbf{X}) = \sum_{h=1}^{H} \mathbf{X}_{\text{attn}}^h \mathbf{W}_O^{h,:,:} = \sum_{h=1}^{H} (\text{Softmax}((\mathbf{X}\mathbf{W}_Q^{:,h,:})(\mathbf{X}\mathbf{W}_K^{:,h,:})^\top)\mathbf{X}\mathbf{W}_V^{:,h,:})\mathbf{W}_O^{h,:,:}. \quad (1)$$

An MLP block contains two linear layers with weights $\mathbf{W}_1 \in \mathbb{R}^{C \times D}$ and $\mathbf{W}_2 \in \mathbb{R}^{D \times C}$, where $D$ represents the hidden channels, set as $4C$ by default in DiT. To dynamically control the MLP width, we divide $D$ hidden channels into $H$ groups, reformulating the weights into $\mathbf{W}_1 \in \mathbb{R}^{C \times (H \times D_H)}$ and $\mathbf{W}_2 \in \mathbb{R}^{(H \times D_H) \times C}$, where $D_H = D/H$. Hence, the operation in MLP can be formulated as:

$$\text{MLP}(\mathbf{X}) = \sum_{h=1}^{H} \sigma(\mathbf{X}_{\text{hidden}}^h)\mathbf{W}_2^{h,:,:} = \sum_{h=1}^{H} \sigma(\mathbf{X}\mathbf{W}_1^{:,h,:})\mathbf{W}_2^{h,:,:}, \quad (2)$$

where $\sigma$ denotes the activation layer.

**Dynamic width control based on timestep.** To dynamically activate the heads and channel groups at each diffusion timestep, in each block, we feed the timestep embedding $\mathbf{E}_t \in \mathbb{R}^C$ into routers $\text{R}_{\text{head}}$ and $\text{R}_{\text{channel}}$ (Figure 2(a)). Each router comprises a linear layer followed by the Sigmoid function, producing the probability of each head and channel group to be activated:

$$\mathbf{S}_{\text{head}} = \text{R}_{\text{head}}(\mathbf{E}_t) \in [0,1]^H, \quad \mathbf{S}_{\text{channel}} = \text{R}_{\text{channel}}(\mathbf{E}_t) \in [0,1]^H. \quad (3)$$

A threshold of 0.5 is then used to convert the continuous-valued $\mathbf{S}_{\text{head}}$ and $\mathbf{S}_{\text{channel}}$ into binary masks $\mathbf{M}_{\text{head}} \in \{0,1\}^H$ and $\mathbf{M}_{\text{channel}} \in \{0,1\}^H$, indicating the activation decisions for attention heads and channel groups. The $h$-th head (group) is activated only when $\mathbf{M}_{\text{head}}^h = 1$ ($\mathbf{M}_{\text{channel}}^h = 1$). Benefiting from the grouping operation, routers introduce negligible parameters and computation.

**Inference.** After obtaining the discrete decisions $\mathbf{M}_{\text{head}}$ and $\mathbf{M}_{\text{channel}}$, each DyDiT block only computes the activated heads and channel groups during generation:

$$\begin{aligned}
\text{MHSA}(\mathbf{X}) &= \sum_{h:\mathbf{M}_{\text{head}}^h=1} \mathbf{X}_{\text{attn}}^h \mathbf{W}_O^{h,:,:}, \\
\text{MLP}(\mathbf{X}) &= \sum_{h:\mathbf{M}_{\text{channel}}^h=1} \sigma(\mathbf{X}_{\text{hidden}}^h)\mathbf{W}_2^{h,:,:}.
\end{aligned} \quad (4)$$

Let $\tilde{H}_{\text{head}} = \sum_h \mathbf{M}_{\text{head}}^h$ and $\tilde{H}_{\text{channel}} = \sum_h \mathbf{M}_{\text{channel}}^h$ denote the number of activated heads/groups. TWD reduces the MHSA computation from $\mathcal{O}(H \times (4NCC_H + 2N^2C_H))$ to $\mathcal{O}(\tilde{H}_{\text{head}} \times (4NCC_H + 2N^2C_H))$ and MLP blocks from $\mathcal{O}(H \times 2NCD_H)$ to $\mathcal{O}(\tilde{H}_{\text{channel}} \times 2NCD_H)$. It is worth noting

that as the activation choices depend solely on the timestep $\mathbf{E}_t$, we can pre-compute the masks offline once the training is completed, and *pre-define* the activated network architecture before deployment. This avoids the sample-dependent inference graph in traditional dynamic architectures (Meng et al., 2022; Han et al., 2024c) and facilitates the realistic speedup in batched inference.

### 3.3 SPATIAL-WISE DYNAMIC TOKEN

In addition to the timestep dimension, the redundancy widely exists in the spatial dimension due to the varying complexity of different patches (Figure 1(b)). To this end, we propose a spatial-wise dynamic token (SDT) method to reduce computation for the patches where noise estimation is "easy".

**Bypassing the MLP block.** As shown in Figure 2 (b), SDT adaptively identifies the tokens associated with image regions that present lower noise prediction difficulty. These tokens are then allowed to bypass the computationally intensive MLP blocks. Theoretically, this block-bypassing operation can be applied to both MHSA and MLP. However, we find MHSA crucial for establishing token interactions, which is essential for the generation quality. More critically, varying token numbers across images in MHSA could result in incomplete tensor shapes in a batch, reducing the overall throughput in generation. Therefore, SDT is applied only in MLP blocks in each layer.

Concretely, before each MLP block, we feed the input $\mathbf{X} \in \mathbb{R}^{N \times C}$ into a token router $\mathrm{R}_{\mathrm{token}}$, which predicts the probability $\mathbf{S}_{\mathrm{token}} \in \mathbb{R}^N$ of each token to be processed. This can be formulated as:

$$\mathbf{S}_{\mathrm{token}} = \mathrm{R}_{\mathrm{token}}(\mathbf{X}) \in [0, 1]^N. \tag{5}$$

We then convert it into a binary mask $\mathbf{M}_{\mathrm{token}}$ using a threshold of 0.5. Each element $\mathbf{M}_{\mathrm{token}}^i \in \{0, 1\}$ in the mask indicates whether the $i$-th token should be processed by the block (if $\mathbf{M}_{\mathrm{token}}^i = 1$) or directly bypassed (if $\mathbf{M}_{\mathrm{token}}^i = 0$). The router parameters are not shared across different layers.

**Inference.** During inference (Figure 2(b)), we gather the tokens based on the mask $\mathbf{M}_{\mathrm{token}}$ and feed them to the MLP, thereby avoiding unnecessary computational costs for other tokens. Then, we adopt a scatter operation to reposition the processed tokens. This further reduces the computational cost of the MLP block from $\mathcal{O}(\tilde{H}_{\mathrm{channel}} N \times 2CD_H)$ to $\mathcal{O}(\tilde{H}_{\mathrm{channel}} \tilde{N} \times 2CD_H)$, where $\tilde{N} = \sum_i \mathbf{M}_{\mathrm{token}}^i$ denotes the actual number of tokens to be processed. Since there is no token interaction within the MLP, the SDT operation supports batched inference, improving the practical generation efficiency.

### 3.4 FLOPS-AWARE END-TO-END TRAINING

In the following, we first present the details of end-to-end training, followed by the loss design for controlling the computational complexity of DyDiT and techniques to stabilize fine-tuning.

**End-to-end training.** During training, in TWD, we multiply $\mathbf{M}_{\mathrm{head}}$ and $\mathbf{M}_{\mathrm{channel}}$ with their corresponding features ($\mathbf{X}_{\mathrm{attn}}$ and $\mathbf{X}_{\mathrm{hidden}}$) to zero out the deactivated heads and channel groups, respectively. Similarly, in SDT, we multiply $\mathbf{M}_{\mathrm{token}}$ with $\mathrm{MLP}(\mathbf{X})$ to deactivate the tokens that should not be processed by MLP. Straight-through-estimator (Bengio et al., 2013) and Gumbel-Sigmoid (Meng et al., 2022) are employed to enable the end-to-end training of routers.

**Training with FLOPs-constrained loss.** We design a FLOPs-constrained loss to control the computational cost during the generation process. We find it impractical to obtain the entire computation graph during $T$ timesteps since the total timestep $T$ is large *e.g.* $T = 1000$. Fortunately, the timesteps in a batch are sampled from $t \sim \mathrm{Uniform}(0, T)$ during training, which approximately covers the entire computation graph. Let $B$ denote the batch size, with $t_b$ as the timestep for the $b$-th sample, we compute the total FLOPs at the sampled timestep, $F_{\mathrm{dynamic}}^{t_b}$, using masks $\{\mathbf{M}_{\mathrm{head}}^{t_b}, \mathbf{M}_{\mathrm{channel}}^{t_b}, \mathbf{M}_{\mathrm{token}}^{t_b}\}$ from each transformer layer, as detailed in Section 3.2 and Section 3.3. Let $F_{\mathrm{static}}$ denote the total FLOPs of MHSA and MLP blocks in the static DiT. We formulate the FLOPs-constrained loss as:

$$\mathcal{L}_{\mathrm{FLOPs}} = \left( \frac{1}{B} \sum_{t_b : b \in [1, B]} \frac{F_{\mathrm{dynamic}}^{t_b}}{F_{\mathrm{static}}} - \lambda \right)^2, \tag{6}$$

where $\lambda$ is a hyperparameter representing the target FLOPs ratio, and $t_b$ is uniformly sampled from the interval $[0, T]$. The overall training objective combines this FLOPs-constrained loss with the original DiT training loss, expressed as $\mathcal{L} = \mathcal{L}_{\mathrm{DiT}} + \mathcal{L}_{\mathrm{FLOPs}}$.

**Fine-tuning stabilization.** In practice, we find directly finetuning DyDiT with $\mathcal{L}$ might occasionally lead to unstable training. To address this, we employ two stabilization techniques. First, for a warm-up phase we maintain a complete DiT model supervised by the same diffusion target, introducing an

Table 1: **Comparison with diffusion models on ImageNet of 256×256 and 512×512 resolutions.** DyDiT-XL achieves competitive performance while significantly reducing the computational cost.

| Model | Params. (M) ↓ | FLOPs (G) ↓ | FID ↓ | sFID ↓ | IS ↑ | Precision ↑ | Recall ↑ |
|---|---|---|---|---|---|---|---|
| *Static* $256 \times 256$ | | | | | | | |
| ADM | 608 | 1120 | 4.59 | 5.25 | 186.87 | 0.82 | 0.52 |
| LDM-4 | 400 | 104 | 3.95 | - | 178.22 | 0.81 | 0.55 |
| U-ViT-L/2 | **287** | 77 | 3.52 | - | - | - | - |
| U-ViT-H/2 | 501 | 113 | 2.29 | - | 247.67 | **0.87** | 0.48 |
| DiffuSSM-XL | 673 | 280 | 2.28 | **4.49** | 269.13 | 0.86 | 0.57 |
| DiM-L | 380 | 94 | 2.64 | - | - | - | - |
| DiM-H | 860 | 210 | 2.21 | - | - | - | - |
| DiT-L | 468 | 81 | 5.02 | - | 167.20 | 0.75 | 0.57 |
| DiT-XL | 675 | 118 | 2.27 | 4.60 | 277.00 | 0.83 | 0.57 |
| DiffiT | 561 | 114 | 1.73 | - | 276.49 | 0.80 | 0.62 |
| SiT-XL | 675 | 118 | 2.06 | **4.49** | 277.50 | 0.83 | 0.59 |
| DiMR-XL | 505 | 160 | **1.70** | - | **289.00** | 0.79 | **0.63** |
| *Dynamic* $256 \times 256$ | | | | | | | |
| DyDiT-XL$_{\lambda=0.7}$ | 678 | 84.33 | 2.12 | 4.61 | 284.31 | 0.81 | 0.60 |
| DyDiT-XL$_{\lambda=0.5}$ | 678 | **57.88** | 2.07 | 4.56 | 248.03 | 0.80 | 0.61 |
| *Static* $512 \times 512$ | | | | | | | |
| DiT-XL | **675** | 514 | 3.04 | 5.02 | 240.80 | 0.84 | 0.54 |
| ADM-G | 731 | 2813 | 3.85 | 5.86 | 221.72 | 0.84 | 0.53 |
| DiffuSSM-XL | 673 | 1066 | 3.41 | - | **255.00** | **0.85** | 0.49 |
| DiM-Huge | 860 | 708 | 3.78 | - | - | - | - |
| SiT-XL | **675** | 514 | **2.62** | **4.18** | 252.21 | 0.84 | **0.57** |
| *Dynamic* $512 \times 512$ | | | | | | | |
| DyDiT-XL$_{\lambda=0.7}$ | 678 | **375.05** | 2.88 | 5.14 | 228.93 | 0.83 | 0.56 |

additional item, $\mathcal{L}_{\text{DiT}}^{\text{complete}}$ along with $\mathcal{L}$. After this phase, we remove this item and continue training solely with $\mathcal{L}$. Additionally, prior to fine-tuning, we rank the heads and hidden channels in MHSA and MLP blocks based on a magnitude criterion (He et al., 2017). We consistently select the most important head and channel group in TDW. This ensures that at least one head and channel group is activated in each MHSA and MLP block across all timesteps, thereby alleviating the instability.

# 4 EXPERIMENTS

**Implementation details.** Our DyDiT can be built easily by fine-tuning on pre-trained DiT weights. We experiment on three different-sized DiT models denoted as DiT-S/B/XL. For DiT-XL, we directly adopt the checkpoint from the official DiT repository (Peebles & Xie, 2023), while for DiT-S and DiT-B, we use pre-trained models provided in Pan et al. (2024). All experiments are conducted on a server with 8×NVIDIA A800 80G GPUs. More details of model configurations and training setup can be found in Appendix A.1 and A.2, respectively. Following DiT (Peebles & Xie, 2023), the strength of classifier-free guidance (Ho & Salimans, 2022) is set to 1.5 and 4.0 for evaluation and visualization, respectively. Unless otherwise specified, 250 DDPM (Ho et al., 2020) sampling steps are used. All speed tests are performed on an NVIDIA V100 32G GPU.

**Datasets.** Following the protocol in DiT (Peebles & Xie, 2023), we mainly conduct experiments on ImageNet (Deng et al., 2009) at a resolution of $256 \times 256$. To comprehensively evaluate our method, we also assess performance and efficiency on four fine-grained datasets used by Xie et al. (2023): Food (Bossard et al., 2014), Artbench (Liao et al., 2022), Cars (Gebru et al., 2017) and Birds (Wah et al., 2011). We conduct experiments in both in-domain fine-tuning and cross-domain transfer learning manners on these dataset. Images of these datasets are also resized into $256 \times 256$ resolution.

**Metrics.** Following prior works (Peebles & Xie, 2023; Teng et al., 2024), we sample 50,000 images to measure the Fréchet Inception Distance (FID) (Heusel et al., 2017) score with the ADM's TensorFlow evaluation suite (Dhariwal & Nichol, 2021). Inception Score (IS) (Salimans et al., 2016), sFID (Nash et al., 2021), and Prevision-Recall (Kynkäänniemi et al., 2019) are also reported for complementary. **Bold font** and underline denote the best and the second-best performance, respectively.

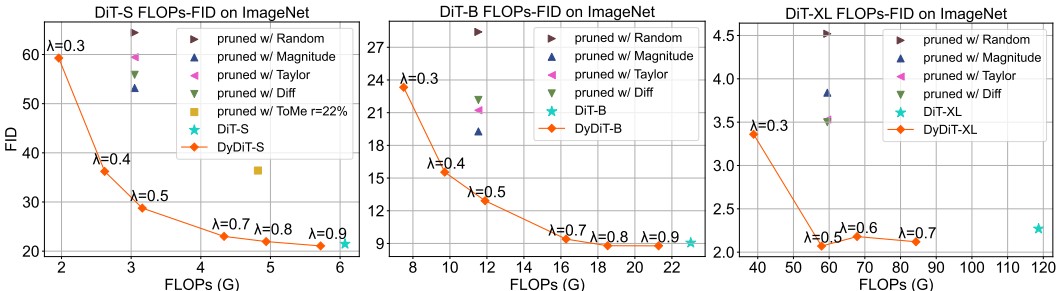

Figure 3: **FLOPs-FID trade-off for S, B, and XL size models on ImageNet.** For clarity, we omit the results of applying ToMe to DiT-B and DiT-XL, as it does not surpass the random pruning.

## 4.1 COMPARISON WITH STATE-OF-THE-ART DIFFUSION MODELS

In Table 1, we compare our method with other representative diffusion models, including ADM (Dhariwal & Nichol, 2021), LDM (Rombach et al., 2022), U-ViT (Bao et al., 2023), DiffuSSM (Yan et al., 2024), DiM (Teng et al., 2024), and DiT (Peebles & Xie, 2023), SiT (Ma et al., 2024a), DiffiT (Hatamizadeh et al., 2025), DiMR (Liu et al., 2024) on ImageNet generation. All methods except ours adopt a static architecture. DyDiT-XL is fine-tuned with fewer than 3% additional iterations based on DiT to adapt the dynamic architecture, as detailed in Appendix A.1.

Notably, our model DyDiT$_{\lambda=0.5}$ achieves a 2.07 FID score with less than 50% FLOPs of its counterpart, DiT-XL, and outperforms most models obviously. This verify that our method can effectively remove the redundant computation in DiT and maintain the generation performance. With around 80G FLOPs, our DyDiT$_{\lambda=0.7}$ method significantly outperforms U-ViT-L/2 and DiT-L, further validing the advantages of our dynamic generation paradigm. Under the $512\times512$ resolution, our method can also achieve performance comparable to SiT-XL with significantly fewer FLOPs.

## 4.2 COMPARISON WITH PRUNING METHODS

**Benchmarks.** The proposed timestep-wise dynamic width and spatial-wise dynamic token improve efficiency from the model *architecture* and *token* redundancy perspective, respectively. To evaluate the superiority of our approach, we compare our methods against representative *static* structure and token pruning techniques. More details of this experiment can be found in Appendix A.3.

*Pruning-based methods.* We include Diff pruning Fang et al. (2024) in the comparison, which is a Taylor-based (Molchanov et al., 2016) pruning method specifically optimized for the diffusion process and has demonstrated superiority on diffusion models with U-Net (Ronneberger et al., 2015) architecture (Fang et al., 2024). Following Fang et al. (2024), we also include Random pruning, Magnitude pruning (He et al., 2017), and Taylor pruning (Molchanov et al., 2016) in the comparison. We adopt these four pruning approaches to distinguish important heads and channels in DiT from less significant ones, which can be removed to reduce the model width.

*Token merging.* We also compare our methods with a training-free token pruning technique, ToMe (Bolya et al., 2022), which progressively prunes tokens in each vision transformer (Dosovitskiy et al., 2020) layer through adaptive token merging. Its enhanced version (Bolya & Hoffman, 2023) can also accelerate diffusion models based on U-Net architectures *e.g.* Stable Diffusion Rombach et al. (2022). We directly apply the enhanced version in each layer of DiT.

**Results.** We present the FLOPs-FID curves for S, B, and XL size models in Figure 3. Across differense sizes, DyDiT significantly outperforms all pruning methods with similar or even lower FLOPs, highlighting the superiority of dynamic architecture over static pruning in diffusion transformers.

Interestingly, Magnitude pruning shows slightly better performance among structural pruning techniques on DiT-S and DiT-B, while Diff pruning and Taylor pruning perform better on DiT-XL. This indicates that different-sized DiT prefer distinct pruning criteria. Although ToMe (Bolya & Hoffman, 2023) successfully accelerates U-Net models with acceptable performance loss, its application to DiT results in performance degradation, as also observed in Moon et al. (2023). We conjecture that the errors introduced by token merging become irrecoverable in DiT due to the absence of convolutional layers and long-range skip connections present in U-Net architectures.

**Scaling up ability.** We can observe from Figure 3 that the performance gap between DyDiT and DiT diminishes as model size increases. Specifically, DyDiT-S achieves a comparable FID to the

Table 2: **Results on fine-grained datasets.** The model marked with † corresponds to fine-tuning directly on the target dataset. See the main texts for details.

| Model | s/image ↓ | FLOPs (G) ↓ | FID ↓ | | | | |
| --- | --- | --- | --- | --- | --- | --- | --- |
| | | | Food | Artbench | Cars | Birds | #Average |
| DiT-S | 0.65 | 6.07 | 14.56 | **17.54** | **9.30** | **7.69** | **12.27** |
| pruned w/ random | 0.38 | 3.05 | 45.66 | 76.75 | 60.26 | 48.60 | 57.81 |
| pruned w/ magnitude | 0.38 | 3.05 | 41.93 | 42.04 | 31.49 | 26.45 | 35.44 |
| pruned w/ taylor | 0.38 | 3.05 | 47.26 | 74.21 | 27.19 | 22.33 | 42.74 |
| pruned w/ diff | 0.38 | 3.05 | 36.93 | 68.18 | 26.23 | 23.05 | 38.59 |
| pruned w/ ToMe 20% | 0.61 | 4.82 | 43.87 | 62.96 | 32.16 | 15.20 | 38.54 |
| DyDiT-S$_{\lambda=0.5}$ | 0.41 | 3.16 | 16.74 | 21.35 | 10.01 | 7.85 | 13.98 |
| DyDiT-S$_{\lambda=0.5}$† | 0.41 | 3.17 | **13.03** | 19.47 | 12.15 | 8.01 | 13.16 |

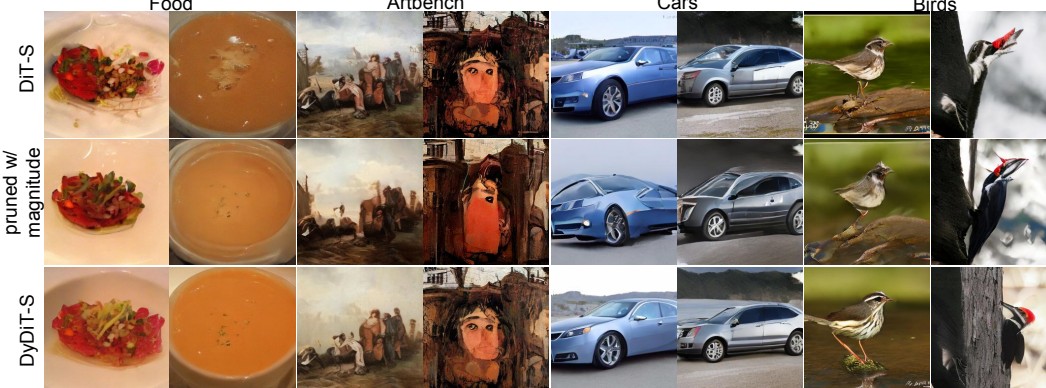

Figure 4: **Qualitative comparison of images generated by the original DiT, DiT pruned with magnitude, and DyDiT.** All models are of "S" size. The FLOPs ratio $\lambda$ in DyDiT is set to 0.5.

original DiT only at $\lambda = 0.9$, while DyDiT-B achieves this with a lower FLOPs ratio, e.g., $\lambda = 0.7$. When scaled to XL, DyDiT-XL attains a slightly better FID even at $\lambda = 0.5$. This is due to increased computation redundancy with larger models, allowing our method to reduce redundancy without compromising FID. These results validate the scalability of our approach, which is crucial in the era of large models, encouraging further exploration of larger models in the future.

## 4.3 RESULTS ON FINE-GRAINED DATASETS

**Quantitative results.** We further compare our method with structural pruning and token pruning approaches on fine-grained datasets under the in-domain fine-tuning setting, where the DiT is initially pre-trained on the corresponding dataset and subsequently fine-tuned on the same dataset for pruning or dynamic adaptation. Detailed experiment settings are presented in Appendix A.4. Results are summarized in Table 2. With the pre-defined FLOPs ratio $\lambda = 0.5$, our method significantly reduces computational cost and enhances generation speed while maintaining performance levels comparable to the original DiT. To ensure fair comparisons, we set width pruning ratios to 50% for pruning methods, aiming for similar FLOPs. Among structural pruning techniques, Magnitude pruning shows relatively better performance, yet DyDiT consistently outperforms it by a substantial margin. With a 20% merging ratio, ToMe also speeds up generation but sacrifices performance. As mentioned, the lack of convolutional layers and skip connections makes applying ToMe to DiT suboptimal.

**Qualitative visualization.** Figure 4 presents images generated by DyDiT-S on fine-grained datasets, compared to those produced by the original or pruned DiT-S. These qualitative results demonstrate that our method maintains the FID score while producing images of quality comparable to DiT-S.

**Cross-domain transfer learning.** Transferring to downstream datasets is a common practice to leverage pre-trained generations models. In this experiment, we fine-tune a model pre-trained on ImageNet to perform cross-domain adaptation on the target dataset while concurrently learning the dynamic architecture, yielding DyDiT-S$_{\lambda=0.5}$† in Table 2. More details are presented in Appendix A.5. We can observe that learning the dynamic architecture during the cross domain transfer learning does not hurt the performance, and even leads to slight better average FID score than DyDiT-S$_{\lambda=0.5}$. This further broadens the application scope of our method.

Table 3: **Ablation Study** on DyDiT-S$_{\lambda=0.5}$. All models evoke around 3.16 GFLOPs.

| Model | TDW | SDT | FID ↓ | | | | | |
|---|---|---|---|---|---|---|---|---|
| | | | ImageNet | Food | Artbench | Cars | Birds | #Average |
| I | ✓ | | 31.89 | **15.71** | 28.19 | 19.67 | 9.23 | 20.93 |
| II | | ✓ | 70.06 | 23.79 | 52.78 | 16.90 | 12.05 | 35.12 |
| III | ✓ | ✓ | **28.75** | 16.74 | **21.35** | **10.01** | **7.85** | **16.94** |
| I (random) | | | 124.38 | 111.88 | 151.99 | 127.53 | 164.29 | 136.01 |
| I (manual) | | | 34.08 | 23.89 | 40.02 | 22.34 | 20.17 | 28.10 |
| III (layer-skip) | ✓ | | 30.95 | 17.75 | 23.15 | 10.53 | 9.01 | 18.29 |

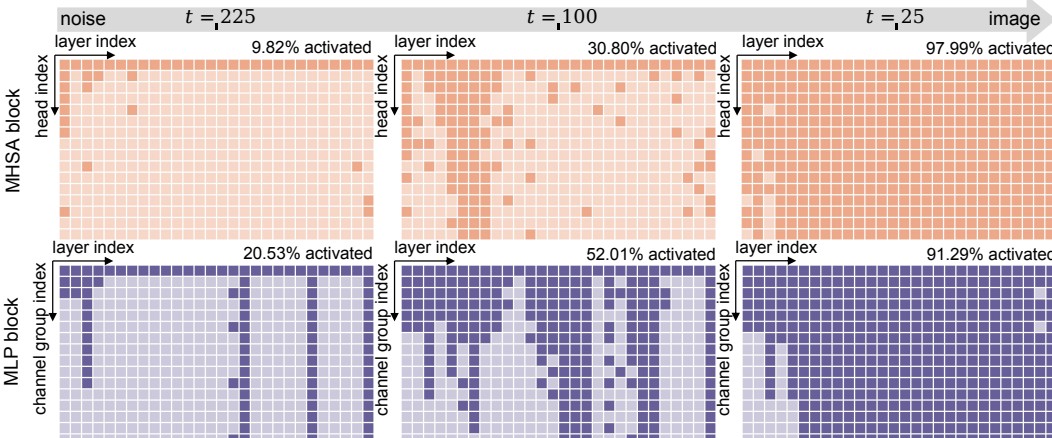

Figure 5: **Visualization of dynamic architecture.** ▨ and ▧ indicates the deactivated and activated heads in an MHSA block, while ▨ and ▧ denotes that the channel group is deactivated or activated in an MLP block, respectively. We conduct 250-step DDPM generation.

## 4.4 ABLATION STUDY

**Main components.** We first conduct experiments to verify the effectiveness of each component in our method. We summarize the results in Table 3. "I" and "II" denote DiT with only the proposed timstep-wise dynamic width (TDW) and spatial-wise dynamic token (SDT), respectively. We can find that "I" performs much better than "II". This is attributed to the fact that, with the target FLOPs ratio $\lambda$ set to 0.5, most tokens in "II" have to bypass MLP blocks, leaving only MHSA blocks to process tokens, significantly affecting performance (Dong et al., 2021). "III" represents the default model that combines both TDW and SDT, achieving obviously better performance than "I" and "II". Given a computational budget, the combination of TDW and SDT allows the model to discover computational redundancy from both the time-step and spatial perspectives.

**Importance of routers in temporal-wise dynamic width.** Routers in TDW adaptively adjust the model width for each block across all timesteps. Replacing the learnable router with a random selection, resulting in "I (random)", leads to model collapse across all datasets. This is due to the random activation of heads and channel groups, which hinders the model's ability to generate high-quality images. We also experiment a manually-designed strategy, termed "I (manual)", in which we activate $5/6$, $1/2$, $1/3$, $1/3$ of the heads and channels for the intervals $[0, 1/T]$, $[1/T, 2/T]$, $[2/T, 3/4T]$, and $[3/4T, T]$ timesteps, respectively. This results in around 50% average FLOPs reduction. Since this strategy aligns the observation in Figure 1(a) and allocates more computation to timesteps approaching 0, "I (manual)" outperforms "I (random)" obviously. However, it does not surpass "I", highlighting the importance of learned routers.

**Importance of token-level bypassing in spatial-wise dynamic token.** We also explore an alternative design to conduct token bypassing. Specifically, each MLP block adopts a router to determine whether all tokens of an image should bypass the block. This modification causes SDT to become a layer-skipping approach (Wang et al., 2018). We replace SDT in "III" with this design, resulting in "III (layer-skip)" in Table 3. As outlined in Section 1, varying regions of an image face distinct challenges in noise prediction. A uniform token processing strategy fails to address this heterogeneity effectively. For example, tokens from complex regions might bypass essential blocks, resulting in suboptimal noise prediction. The results presented in Table 3 further confirm that the token-level bypassing in SDT, obviously improves the performance of "III" compared to "III (layer-skip)".

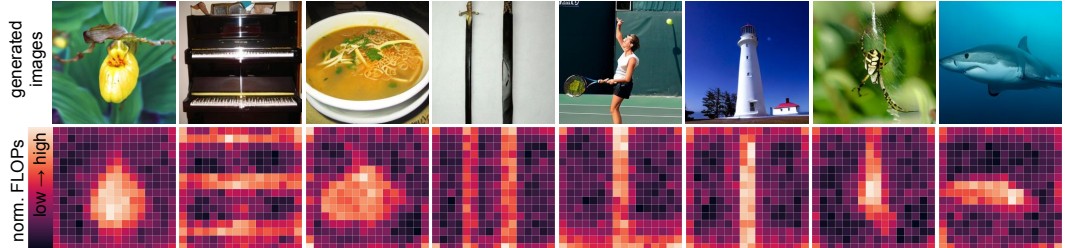

Figure 6: **Computational cost across different image patches.** We quantify the FLOPs cost on image patches over the generation process and normalize them into [0, 1] for better clarity.

Table 4: **Combination with efficient samplers** (Song et al., 2020a; Lu et al., 2022).

| Model | 250-DDPM | | 50-DDIM | | 20-DPM-solver++ | | 10-DPM-solver++ | |
|---|---|---|---|---|---|---|---|---|
| | s/image ↓ | FID ↓ | s/image ↓ | FID ↓ | s/image ↓ | FID ↓ | s/image ↓ | FID ↓ |
| DiT-XL | 10.22 | 2.27 | 2.00 | 2.26 | 0.84 | 4.62 | 0.42 | 11.66 |
| DyDiT-XL$_{\lambda=0.7}$ | 7.76 | 2.12 | 1.56 | 2.16 | 0.62 | 4.28 | 0.31 | 11.10 |
| DyDiT-XL$_{\lambda=0.5}$ | 5.91 | 2.07 | 1.17 | 2.36 | 0.46 | 4.22 | 0.23 | 11.31 |

### 4.5 VISUALIZATION

**Learned timestep-wise dynamic strategy.** Figure 5 illustrates the activation patterns of heads and channel groups during the 250-step DDPM generation process. Throughout this process, TWD progressively activates more MHSA heads and MLP channel groups as it transitions from noise to image. As discussed in Section 1, prediction is more straightforward when generation is closer to noise (larger $t$) and becomes increasingly challenging as it approaches the image (smaller $t$). Our visualization corroborates this observation, demonstrating that the model allocates more computational resources to more complex timesteps. Notably, the activation rate of MLP blocks surpasses that of MHSA blocks at $t = 255$ and $t = 100$. This can be attributed to the token bypass operation in the spatial-wise dynamic token (SDT), which reduces the computational load of MLP blocks, enabling TWD to activate additional channel groups with minimal computational overhead.

**Spatial-wise dynamic token adapts computational cost on each image patch.** We quantify and normalize the computational cost on different image patches during generation, ranging from [0, 1] in Figure 6. These results verify that our SDT effectively learns to adjust computational expenditure based on the complexity of image patches. SDT prioritizes challenging patches containing detailed and colorful main objects. Conversely, it allocates less computation to background regions characterized by uniform and continuous colors. This behavior aligns with our findings in Figure 1(b).

### 4.6 COMBINATION WITH EFFICIENT SAMPLERS.

Our DyDiT is a general architecture which can be seamlessly incorporated with efficient samplers such as DDIM (Song et al., 2020a) and DPM-solver++ (Lu et al., 2022). As presented in Table 4, when using the 50-step DDIM, both DiT-XL and DyDiT-XL exhibit significantly faster generation, while our method consistently achieving higher efficiency due to its dynamic computation paradigm. When we further reduce the sampling step to 20 and 10 with DPM-solver++, we observe an FID increasement on all models, while our method still achieves competitive performance compared to the original DiT. These findings highlight the potential of integrating our approach with efficient samplers, suggesting a promising avenue for future research.

## 5 DISCUSSION AND CONCLUSION

In this study, we investigate the training process of the Diffusion Transformer (DiT) and identify significant computational redundancy associated with specific diffusion timesteps and image patches. To this end, we propose Dynamic Diffusion Transformer (DyDiT), an architecture that can adaptively adjust the computation allocation across different timesteps and spatial regions. Comprehensive experiments on various datasets and model sizes validate the effectiveness of DyDiT. We anticipate that the proposed method will advance the development of transformer-based diffusion models.

**Limitations and future works.** Similarly to DiT, the proposed DyDiT is currently focusing on image generation. In future works, DyDiT could be further explored to be applied to other tasks, such as video generation (Ma et al., 2024b) and controllable generation (Chen et al., 2024).

**Acknowledgments.** This work was supported by Damo Academy through Damo Academy Research Intern Program. This work also was supported by the National Research Foundation, Singapore under its AI Singapore Programme (AISG Award No: AISG2-PhD-2021-08-008). Yang You's research group is being sponsored by NUS startup grant (Presidential Young Professorship), Singapore MOE Tier-1 grant, ByteDance grant, ARCTIC grant, SMI grant (WBS number: A8001104-00-00), Alibaba grant, and Google grant for TPU usage.

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

We organize our appendix as follows.

**Experimental settings:**

- Section A.1: Training details of DyDiT on ImageNet.
- Section A.2: Model configurations of both DiT and DyDiT.
- Section A.3: Implement details of pruning methods on ImageNet.
- Section A.4: Details of in-domain fine-tuning on fine-grained datasets.
- Section A.5: Details of cross-domain fine-tuning.

**Additional results:**

- Section B.1: The inference speed of DyDiT and its acceleration over DiT across models of varying sizes and specified FLOP budgets.
- Section B.2: The generalization capability of our method on the U-ViT (Bao et al., 2023) architecture.
- Section B.3: Further fine-tuning the original DiT to show that the competitive performance of our method is not due to the additional fine-tuning.
- Section B.4: The effectiveness of DyDiT on $512 \times 512$ resolution image generation.
- Section B.5: The effectiveness of DyDiT in text-to-image generation, based on PixArt (Chen et al., 2023).
- Section B.6: Integration of DyDiT with a representative distillation-based efficient sampler, the latent consistency model (LCM) (Luo et al., 2023).
- Section B.7: Comparison between DyDiT with the early exiting diffusion model (Moon et al., 2023).
- Section B.8: Fine-tuning efficiency of DyDiT. We fine-tune our model by fewer iterations.
- Section B.9: Data efficiency of DyDiT. Our model is fine-tuned on only 10% of the training data.
- Section B.10: We combined our method with DeepCache (Ma et al., 2023).

**Visualization:**

- Section C.1: Additional visualizations of loss maps of DiT-XL.
- Section C.2: Additional visualizations of computational cost across different image patches.
- Section C.3: Visualization of images generated by DyDiT-XL$_{\lambda=0.5}$ on the ImageNet dataset at at resolution of $256 \times 256$.
- Section C.4: Visualization of DyDiT with different $\lambda$s.
- Section C.5: Visual comparison of images generated by PixArt (Chen et al., 2023) and the proposed DyPixArt on the COCO dataset.

**Others:**

- Section D: Frequently asked questions.

## A EXPERIMENTAL SETTINGS.

### A.1 TRAINING DETAILS OF DYDIT ON IMAGENET

In Table 6, we present the training details of our model on ImageNet. For DiT-XL, which is pre-trained over 7,000,000 iterations, only 200,000 additional fine-tuning iterations (around 3%) are needed to enable the dynamic architecture ($\lambda = 0.5$) with our method. For a higher target FLOPs ratio $\lambda = 0.7$, the iterations can be further reduced.

| model | DiT-S | DiT-B | DiT-XL |
|---|---|---|---|
| optimizer | | AdamW (Loshchilov, 2017), learning rate=1e-4 | |
| global batch size | | 256 | |
| target FLOPs ratio $\lambda$ | [0.9, 0.8, 0.7, 0.5, 0.4, 0.3] | [0.9, 0.8, 0.7, 0.5, 0.4, 0.3] | [0.7, 0.6, 0.5, 0.3] |
| fine-tuning iterations | 50,000 | 100,000 | 150,000 for $\lambda = 0.7$ 200,000 for others |
| warmup iterations | 0 | 0 | 30,000 |
| augmentation | | random flip | |
| cropping size | | 224×224 | |

Table 6: Experimental settings of our adaption framework.

### A.2 DETAILS OF DIT AND DYDIT MODELS

We present the configuration details of the DiT and DyDiT models in Table 7. For DiT-XL, we use the checkpoint from the official DiT repository[1] Pan et al. (2024). For DiT-S and DiT-B, we leverage pre-trained models from a third-party repository[2] provided by Pan et al. (2024).

Table 7: **Details of DiT and DyDiT models.** The router in DyDiT introduce a small number of parameters. † denotes that the architecture is dynamically adjusted during generation.

| model | params. (M) ↓ | layers | heads | channel | pre-training | source |
|---|---|---|---|---|---|---|
| DiT-S | 33 | 12 | 6 | 384 | 5M iter | Pan et al. (2024) |
| DiT-B | 130 | 12 | 12 | 768 | 1.6M iter | Pan et al. (2024) |
| DiT-XL | 675 | 28 | 16 | 1152 | 7M iter | Peebles & Xie (2023) |
| DyDiT-S | 33 | 12 | 6 † | 384 † | - | - |
| DyDiT-B | 131 | 12 | 12 † | 768 † | - | - |
| DyDiT-XL | 678 | 28 | 16 † | 1152 † | - | - |

### A.3 COMPARISON WITH PRUNING METHODS ON IMAGENET.

We compare our method with structure pruning and token pruning methods on ImageNet dataset.

- Random pruning, Magnitude Pruning (He et al., 2017), Taylor Pruning (Molchanov et al., 2016), and Diff Pruning (Fang et al., 2024): We adopt the corresponding pruning strategy to rank the importance of heads in multi-head self-attention blocks and channels in MLP blocks. Then, we prune the least important 50% of heads and channels. The pruned model is then fine-tuned for the same number of iterations as its DyDiT counterparts.

- ToMe (Bolya & Hoffman, 2023): Originally designed to accelerate transformer blocks in the U-Net architecture, ToMe operates by merging tokens before the attention block and then unmerging them after the MLP blocks. We set the token merging ratio to 20% in each block.

### A.4 IN-DOMAIN FINE-TUNING ON FINE-TRAINED DATASETS.

We first fine-tune a DiT-S model, which is initialized with parameters pre-trained on ImageNet, on a fine-grained dataset. Following the approach in (Xie et al., 2023), we set the training iteration to

---

[1]https://github.com/facebookresearch/DiT
[2]https://github.com/NVlabs/T-Stitch

24,000. Then, we further fine-tune the model on the same dataset by another 24,000 iterations to adapt the pruning or dynamic architecture to improve the efficiency of the model on the same dataset. We also conduct the generation at a resolution off $224 \times 224$. We search optimal classifier-free guidance weights for these methods.

- Random pruning, Magnitude Pruning (He et al., 2017), Taylor Pruning (Molchanov et al., 2016), and Diff Pruning (Fang et al., 2024): For each method, we rank the importance of heads in multi-head self-attention blocks and channels in MLP blocks, pruning the least important 50%.

- ToMe (Bolya & Hoffman, 2023): Originally designed to accelerate transformer blocks in the U-Net architecture, ToMe operates by merging tokens before the attention block and then unmerging them after the MLP blocks. We set the token merging ratio to 20% in each block.

## A.5 CROSS-DOMAIN TRANSFER LEARNING

In contrast to the aforementioned in-domain fine-tuning, which learns the dynamic strategy within the same dataset, this experiment employs cross-domain fine-tuning. We fine-tune a DiT-S model (pre-trained exclusively on ImageNet) to adapt to the target dataset while simultaneously learning the dynamic architecture. The model is fine-tuned over 48,000 iterations with a batch size of 256.

## B ADDITIONAL RESULTS

### B.1 INFERENCE ACCELERATION.

In Table 8, we present the acceleration ratio of DyDiT compared to the original DiT across different FLOPs targets $\lambda$. The results demonstrate that our method effectively enhances batched inference speed, distinguishing our approach from traditional dynamic networks (Herrmann et al., 2020; Meng et al., 2022; Han et al., 2024c), which adapt inference graphs on a per-sample basis and struggle to improve practical efficiency in batched inference.

Table 8: We conduct batched inference on an NVIDIA V100 32G GPU using the optimal batch size for each model. The actual FLOPs of DyDiT may fluctuate around the target FLOPs ratio.

| model | s/image ↓ | acceleration ↑ | FLOPs (G) ↓ | FID ↓ | FID Δ ↓ |
|---|---|---|---|---|---|
| DiT-S | 0.65 | 1.00 × | 6.07 | 21.46 | +0.00 |
| DyDiT-S$_{\lambda=0.9}$ | 0.63 | 1.03 × | 5.72 | 21.06 | -0.40 |
| DyDiT-S$_{\lambda=0.8}$ | 0.56 | 1.16 × | 4.94 | 21.95 | +0.49 |
| DyDiT-S$_{\lambda=0.7}$ | 0.51 | 1.27 × | 4.34 | 23.01 | +1.55 |
| DyDiT-S$_{\lambda=0.5}$ | 0.42 | 1.54 × | 3.16 | 28.75 | +7.29 |
| DyDiT-S$_{\lambda=0.4}$ | 0.38 | 1.71 × | 2.63 | 36.21 | +14.75 |
| DyDiT-S$_{\lambda=0.3}$ | 0.32 | 2.03 × | 1.96 | 59.28 | +37.83 |
| DiT-B | 2.09 | 1.00 × | 23.02 | 9.07 | +0.00 |
| DyDiT-B$_{\lambda=0.9}$ | 1.97 | 1.05 × | 21.28 | 8.78 | -0.29 |
| DyDiT-B$_{\lambda=0.8}$ | 1.76 | 1.18 × | 18.53 | 8.79 | -0.28 |
| DyDiT-B$_{\lambda=0.7}$ | 1.57 | 1.32 × | 16.28 | 9.40 | +0.33 |
| DyDiT-B$_{\lambda=0.5}$ | 1.22 | 1.70 × | 11.90 | 12.92 | +3.85 |
| DyDiT-B$_{\lambda=0.4}$ | 1.06 | 1.95 × | 9.71 | 15.54 | +6.47 |
| DyDiT-B$_{\lambda=0.3}$ | 0.89 | 2.33 × | 7.51 | 23.34 | +14.27 |
| DiT-XL | 10.22 | 1.00 × | 118.69 | 2.27 | +0.00 |
| DyDiT-XL$_{\lambda=0.7}$ | 7.76 | 1.32 × | 84.33 | 2.12 | -0.15 |
| DyDiT-XL$_{\lambda=0.6}$ | 6.86 | 1.49 × | 67.83 | 2.18 | -0.09 |
| DyDiT-XL$_{\lambda=0.5}$ | 5.91 | 1.73 × | 57.88 | 2.07 | -0.20 |
| DyDiT-XL$_{\lambda=0.3}$ | 4.26 | 2.40 × | 38.85 | 3.36 | +1.09 |

## B.2 EFFECTIVENESS ON U-VIT.

We evaluate the architecture generalization capability of our method through experiments on U-ViT (Bao et al., 2023), a transformer-based diffusion model with skip connections similar to U-Net (Ronneberger et al., 2015). The results, shown in Table 9, indicate that configuring the target FLOPs ratio $\lambda$ to 0.4 and adapting U-ViT-S/2 to our dynamic architecture (denoted as DyUViT-S/2 $_{\lambda=0.4}$) reduces computational cost from 11.34 GFLOPs to 4.73 GFLOPs, while maintaining a comparable FID score. We also compare our method with the structure pruning method Diff Pruning (Fang et al., 2024) and sparse pruning methods ASP (Pool & Yu, 2021; Mishra et al., 2021) and SparseDM (Wang et al., 2024). The results verify the superiority of our dynamic architecture over static pruning.

In Table 10, we apply our method to the largest model, U-ViT-H/2, and conduct experiments on ImageNet. The results demonstrate that our method effectively accelerates U-ViT-H/2 with only a marginal performance drop. These results verify the generalizability of our method in U-ViT.

Table 9: **U-ViT (Bao et al., 2023) performs image generation on the CIFAR-10 dataset (Krizhevsky et al., 2009).** Aligning with its default configuration, we generate images using 1,000 diffusion steps with the Euler-Maruyama SDE sampler (Song et al., 2020b).

| model | s/image ↓ | acceleration ↑ | FLOPs (G) ↓ | FID ↓ | FID Δ ↓ |
|---|---|---|---|---|---|
| U-ViT-S/2 | 2.19 | 1.00 × | 11.34 | 3.12 | 0.00 |
| DyU-ViT-S/2$_{\lambda=0.4}$ | 1.04 | 2.10 × | 4.73 | 3.18 | +0.06 |
| pruned w/ Diff | - | - | 5.32 | 12.63 | +9.51 |
| pruned w/ ASP | - | - | 5.76 | 319.87 | +316.75 |
| pruned w/ SparseDM | - | - | 5.67 | 4.23 | +1.11 |

Table 10: **U-ViT (Bao et al., 2023) performs image generation on the ImageNet (Deng et al., 2009).** Aligning with its default configuration, we generate images using 50-step DPM-solver++(Lu et al., 2022).

| model | s/image ↓ | acceleration ↑ | FLOPs (G) ↓ | FID ↓ | FID Δ ↓ |
|---|---|---|---|---|---|
| U-ViT-H/2 | 2.22 | 1.00 × | 113.00 | 2.29 | 0.00 |
| DyU-ViT-H/2$_{\lambda=0.5}$ | 1.35 | 1.57 × | 67.09 | 2.42 | +0.13 |

## B.3 FURTHER FINE-TUNE ORIGINAL DIT ON IMAGENET.

Our method is not attributed to additional fine-tuning. In Table 11, we fine-tune the original DiT for 150,000 and 350,000 iterations, observing a slight improvement in the FID score, which fluctuates around 2.16. "DiT-XL′" denotes that we introduce the same routers in DiT-XL to maintain the same parameters as that of DyDiT. Under the same iterations, DyDiT achieves a better FID while significantly reducing FLOPs, verifying that the improvement is due to our design rather than extended training iterations.

Table 11: **Further fine-tuneing original DiT on ImageNet.**

| model | pre-trained iterations | fine-tuning iterations | FLOPs (G) ↓ | FID ↓ | FID Δ ↓ |
|---|---|---|---|---|---|
| DiT-XL | 7,000,000 | - | 118.69 | 2.27 | +0.00 |
| DiT-XL | 7,000,000 | 150,000 (2.14%) | 118.69 | 2.16 | -0.11 |
| DiT-XL | 7,000,000 | 350,000 (5.00%) | 118.69 | 2.15 | -0.12 |
| DiT-XL′ | 7,000,000 | 150,000 (2.14%) | 118.69 | 2.15 | -0.12 |
| DyDiT-XL$_{\lambda=0.7}$ | 7,000,000 | 150,000 (2.14%) | 84.33 | 2.12 | -0.15 |

## B.4 EFFECTIVENESS IN HIGH-RESOLUTION GENERATION.

We conduct experiments to generate images at a resolution of 512×512 to validate the effectiveness of our method for high-resolution generation. We use the official checkpoint of DiT-XL 512×512 as

the baseline, which is trained on ImageNet (Deng et al., 2009) for 3,000,000 iterations. We fine-tune it for 150,000 iterations to enable its dynamic architecture, denoted as DyDiT-XL 512×512. The target FLOP ratio is set to 0.7. The experimental results, presented in Table 12, demonstrate that our method achieves a superior FID score compared to the original DiT-XL, while requiring fewer FLOPs.

Table 12: **Image generation at 512×512 resolution on ImageNet (Deng et al., 2009)**. We sample 50,000 images and leverage FID to measure the generation quality. We adopt 100 and 250 DDPM steps to generate images. "FLOPs (G)" denotes the average FLOPs in one timestep.

| model | DDPM steps | s/image ↓ | acceleration ↑ | FLOPs (G) ↓ | FID ↓ | FID Δ ↓ |
|---|---|---|---|---|---|---|
| DiT-XL 512×512 | 100 | 18.36 | 1.00× | 514.80 | 3.75 | 0.00 |
| DyDiT-XL 512×512 $_{\lambda=0.7}$ | 100 | 14.00 | 1.31× | 375.35 | 3.61 | -0.14 |
| DiT-XL 512×512 | 250 | 45.90 | 1.00× | 514.80 | 3.04 | 0.00 |
| DyDiT-XL 512×512 $_{\lambda=0.7}$ | 250 | 35.01 | 1.31× | 375.05 | 2.88 | -0.16 |

### B.5 EFFECTIVENESS IN TEXT-TO-IMAGE GENERATION.

We further validate the applicability of our method in text-to-image generation, which is more challenging than the class-to-image generation. We adopt PixArt-$\alpha$ (Chen et al., 2023), a text-to-image generation model built based on DiT (Peebles & Xie, 2023) as the baseline. PixArt-$\alpha$ is pre-trained on extensive private datasets and exhibits superior text-to-image generation capabilities. Our model is initialized using the official PixArt-$\alpha$ checkpoint fine-tuned on the COCO dataset (Lin et al., 2014). We further fine-tune it with our method to enable dynamic architecture adaptation, resulting in the DyPixArt-$\alpha$ model, as shown in Table 13. Notably, DyPixArt-$\alpha$ with $\lambda = 0.7$ achieves an FID score comparable to the original PixArt-$\alpha$, while significantly accelerating the generation.

Table 13: **Text-to-image generation on COCO (Lin et al., 2014)**. We randomly select text prompts from COCO and adopt 20-step DPM-solver++ (Lu et al., 2022) to sample 30,000 images for evaluating the FID score.

| Model | s/image ↓ | acceleration ↑ | FLOPs (G) ↓ | FID ↓ | FID Δ ↓ |
|---|---|---|---|---|---|
| PixArt-$\alpha$ | 0.91 | 1.00 × | 141.09 | 19.88 | +0.00 |
| DyPixArt-$\alpha_{\lambda=0.7}$ | 0.69 | 1.32 × | 112.44 | 19.75 | -0.13 |

### B.6 EXPLORATION OF COMBINING LCM WITH DYDIT.

Some sampler-based efficient methods (Meng et al., 2023; Song et al., 2023; Luo et al., 2023) adopt distillation techniques to reduce the generation process to several steps. In this section, we combine our DyDiT, a model-based method, with a representative method, the latent consistency model (LCM) (Luo et al., 2023) to explore their compatibility for superior generation speed. In LCM, the generation process can be reduced to 1-4 steps via consistency distillation and the 4-step generation achieves an satisfactory balance between performance and efficiency. Hence, we conduct experiments in the 4-step setting. Under the target FLOPs ratio $\lambda = 0.9$, our method further accelerates generation and achieves comparable performance, demonstrating its potential with LCM. However, further reducing the FLOPs ratio leads to model collapse. This issue may arise because DyDiT's training depends on noise prediction difficulty, which is absent in LCM distillation, causing instability at lower FLOPs ratios. This encourage us to develop dynamic models and training strategies for distillation-based efficient samplers to achieve superior generation efficiency in the future.

### B.7 COMPARISON WITH THE EARLY EXITING METHOD.

We compare our approach with the early exiting diffusion model ASE (Moon et al., 2023; 2024), which implements a strategy to selectively skip layers for certain timesteps. Following their methodology, we evaluate the FID score using 5,000 samples. Results are summarized in Table 15. Despite similar generation performance, our method achieves a better acceleration ratio, demonstrating the effectiveness of our design.

Table 14: **Combining DyDiT with Latent Consistency Model (LCM) (Luo et al., 2023) .** We conduct experiments under the 4-step LCM setting, as it achieves a satisfactory balance between performance and efficiency.

| model | s/image ↓ | FLOPs (G) ↓ | FID ↓ | FID Δ ↓ |
|---|---|---|---|---|
| DiT-XL+250-step DDPM | 10.22 | 118.69 | 2.27 | +0.00 |
| DiT-XL + 4-step LCM | 0.082 | 118.69 | 6.53 | +4.26 |
| DyDiT-XL$_{\lambda=0.9}$ + 4-step LCM | 0.076 | 104.43 | 6.52 | +4.25 |

Table 15: **Comparison with the early exiting method (Moon et al., 2023; 2024).** As methods may be evaluated on different devices, we report only the acceleration ratio for speed comparison.

| model | acceleration ↑ | FID ↓ | FID Δ ↓ |
|---|---|---|---|
| DiT-XL | 1.00 × | 9.08 | 0.00 |
| DyDiT-XL$_{\lambda=0.5}$ | 1.73 × | 8.95 | -0.13 |
| ASE-D4 DiT-XL | 1.34 × | 9.09 | +0.01 |
| ASE-D7 DiT-XL | 1.39 × | 9.39 | +0.31 |

## B.8 TRAINING EFFICIENCY

Our approach enhances the inference efficiency of the diffusion transformer while maintaining training efficiency. It requires only a small number of additional fine-tuning iterations to learn the dynamic architecture. In Table 16, we present our model with various fine-tuning iterations and their corresponding FID scores. The original DiT-XL model is pre-trained on the ImageNet dataset over 7,000,000 iterations with a batch size of 256. Remarkably, our method achieves a 2.12 FID score with just 50,000 fine-tuning iterations to adopt the dynamic architecture-approximately 0.7% of the pre-training schedule. Furthermore, when extended to 100,000 and 150,000 iterations, our method performs comparably to DiT. We observe that the actual FLOPs during generation converge as the number of fine-tuning iterations increases.

Table 16: **Training efficiency.** The original DiT-XL model is pre-trained on the ImageNet dataset over 7,000,000 iterations with a batch size of 256.

| model | fine-tuning iterations | FLOPs (G) ↓ | FID ↓ | FID Δ ↓ |
|---|---|---|---|---|
| DiT-XL | - | 118.69 | 2.27 | +0.00 |
| DyDiT-XL$_{\lambda=0.7}$ | 10,000 (0.14%) | 103.08 | 45.95 | 43.65 |
| DyDiT-XL$_{\lambda=0.7}$ | 25,000 (0.35%) | 91.97 | 2.97 | +0.70 |
| DyDiT-XL$_{\lambda=0.7}$ | 50,000 (0.71%) | 85.07 | 2.12 | -0.15 |
| DyDiT-XL$_{\lambda=0.7}$ | 100,000 (1.43%) | 84.30 | 2.17 | -0.10 |
| DyDiT-XL$_{\lambda=0.7}$ | 150,000 (2.14%) | 84.33 | 2.12 | -0.15 |

## B.9 DATA EFFICIENCY

To evaluate the data efficiency of our method, we randomly sampled 10% of the ImageNet dataset (Deng et al., 2009) for training. DyDiT was fine-tuned on this subset to adapt the dynamic architecture. As shown in Table 17, when fine-tuned on just 10% of the data, our model DyDiT-XL$_{\lambda=0.7}$ still achieves performance comparable to the original DiT. When we further reduce the fine-tuning data ratio to 1%, the FID score increase slightly by 0.06. These results indicate that our method maintains robust performance even with limited fine-tuning data.

## B.10 COMBINATION WITH GLOBAL ACCELERATION.

DeepCache (Ma et al., 2023) is a train-free technique which globally accelerates generation by caching feature maps at specific timesteps and reusing them in subsequent timesteps. As shown in Table 18, with a cache interval of 2, DyDiT achieves further acceleration with only a marginal performance drop. In contrast, DiT with DeepCache requires a longer interval (*e.g.* 5) to achieve

Table 17: **Data efficiency.** The slight difference in FLOPs of our models is introduced by the learned TDW and SDT upon fine-tuning convergence.

| model | fine-tuning data ratio | FLOPs (G) $\downarrow$ | FID $\downarrow$ | FID $\Delta$ $\downarrow$ |
|---|---|---|---|---|
| DiT-XL | - | 118.69 | 2.27 | +0.00 |
| DyDiT-XL$_{\lambda=0.7}$ | 100% | 84.33 | 2.12 | -0.15 |
| DyDiT-XL$_{\lambda=0.7}$ | 10% | 84.43 | 2.13 | -0.14 |
| DyDiT-XL$_{\lambda=0.7}$ | 1% | 84.37 | 2.31 | +0.06 |

Table 18: **Combined with DeepCache.** "interval" denotes the interval of cached timestep in DeepCache (Ma et al., 2023).

| Model | interval | s/image $\downarrow$ | FID $\downarrow$ |
|---|---|---|---|
| DiT-XL | 0 | 10.22 | 2.27 |
| DiT-XL | 2 | 5.02 | 2.47 |
| DiT-XL | 5 | 2.03 | 6.73 |
| DyDiT-XL$_{\lambda=0.5}$ | 0 | 5.91 | 2.08 |
| DyDiT-XL$_{\lambda=0.5}$ | 2 | 2.99 | 2.43 |
| DyDiT-XL$_{\lambda=0.5}$ | 3 | 2.01 | 3.37 |

comparable speed with ours, resulting in an inferior FID score. These results demonstrate the compatibility and effectiveness of our approach in conjunction with DeepCache.

## C   VISUALIZATION

### C.1   ADDITIONAL VISUALIZATION OF LOSS MAPS

In Figure 10, we visualize the loss maps (normalized to the range [0, 1]) for several timesteps, demonstrating that noise in different image patches exhibits varying levels of prediction difficulty.

### C.2   ADDITIONAL VISUALIZATION OF COMPUTATIONAL COST ON IMAGE PATCHES

In Figure 11, we quantify and normalize the computational cost across different image patches during generation, ranging from [0, 1]. The proposed spatial-wise dynamic token strategy learns to adjust the computational cost for each image patch.

### C.3   VISUALIZATION OF SAMPLES FROM DYDIT-XL

We visualize the images generated by DyDiT-XL$_{\lambda=0.5}$ on the ImageNet (Deng et al., 2009) dataset at a resolution of $256 \times 256$ from Figure 12 to Figure 25. The classifier-free guidance scale is set to 4.0. All samples here are uncurated.

### C.4   VISUALIZATION OF DYDIT WITH DIFFERENT $\lambda$

We visualize images generated from DyDiT with different $\lambda$. Images generated from DyDiT-S and DyDiT-XL are presented in Figure 7 and Figure 8, respectively.

For DiT-S and DiT-B, increasing $\lambda$ from 0.3 to 0.7 consistently enhances visual quality. At $\lambda = 0.9$, DyDiT achieves performance on par with the original DiT-S. In the case of DiT-XL, the visual quality of images generated from DyDiT with $\lambda = 0.5$ is comparable to that from the original DiT-XL, attributed to substantial computational redundancy in DiT-XL.

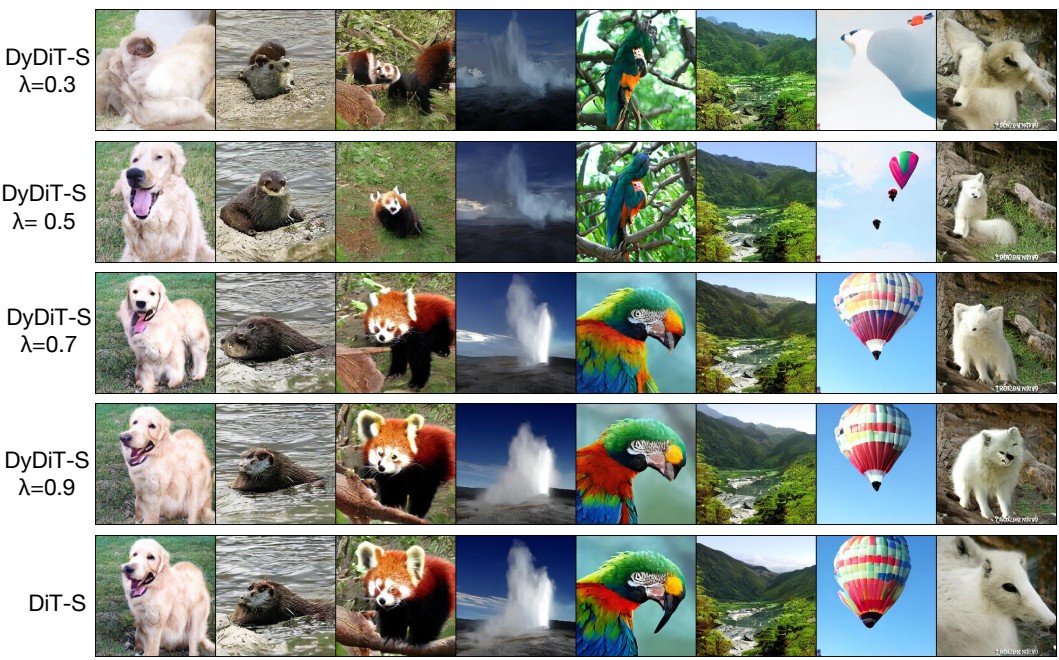

Figure 7: **DyDiT-S.**

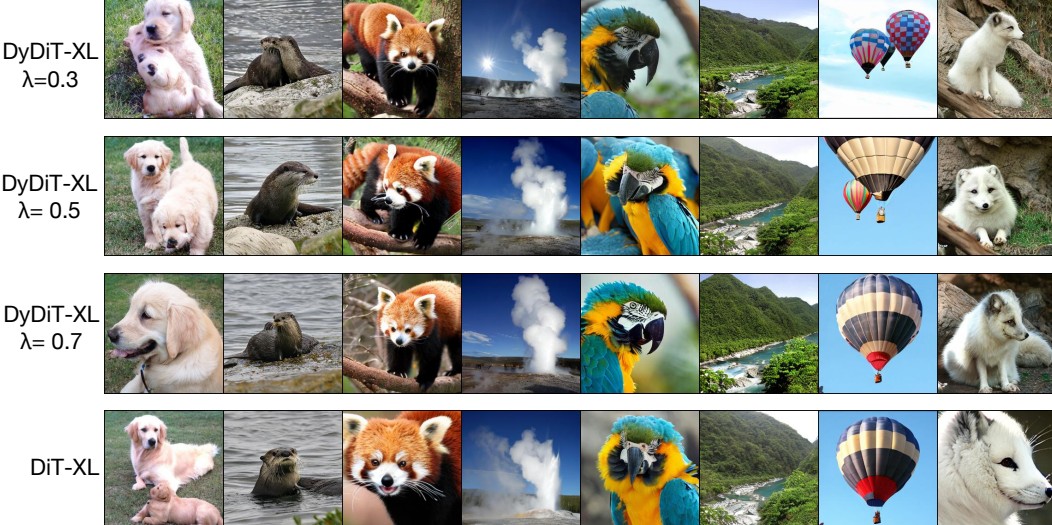

Figure 8: **DyDiT-XL.**

## C.5 VISUALIZATION OF TEXT-TO-IMAGE GENERATION ON COCO

We visualize images generated from the original PixArt-$\alpha$ Chen et al. (2023) and our DyPixArt-$\alpha$ with $\lambda = 0.7$ in Figure 9. The visual quality of images generated from DyPixArt-$\alpha$ is comparable to that from the original PixArt-$\alpha$.

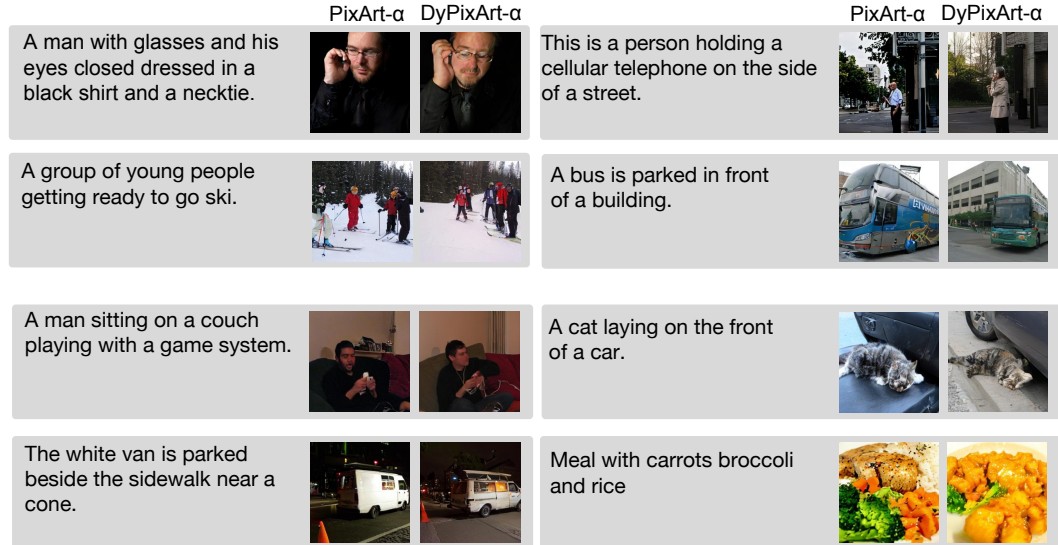

Figure 9: **Visualization from the original PixArt-$\alpha$ and DyPixArt-$\alpha$ with $\lambda = 0.7$.**

## D   FREQUENTLY ASKED QUESTIONS

**Question1: It is unclear how the "pre-define" in L214 benefit the sampling stage?**

Pre-define enables batched inference of our method. The activation of heads and channel groups in TWD relies solely on the timestep $t$, allowing us to pre-calculate activations prior to deployment. By storing the activated indices for each timestep, we can directly access the architecture during generation for a batch of samples. This approach eliminates the sample-dependent inference graph typical in traditional dynamic architectures, enabling efficient and realistic speedup in batched inference.

**Question2: The proposed modules to efficient samplers or to samplers with varying sampling steps remains unclear.**

Consistent with standard practices in samplers such as DDPM, varying the sampling steps translates to differing timestep intervals. We adopt its official code to map $t$ into the range 0–1000, aligning with the 1000 total timesteps used during training. For example, in DDPM with 100 and 250 timesteps:

*a)* 250-DDPM timestep: we map $t \in [249, ....5, 4, 3, 2, 1, 0]$ into $t_{250\text{-DDPM}} \in [999, 995, .....20, 16, 12, 8, 4, 0]$.

*b)* 100-DDPM timestep: we map $t \in [99, 98, ...2, 1, 0]$ into $t_{100\text{-DDPM}} \in [999, 989, ...20, 10, 0]$.

In TWD, we adopt $t_{250\text{-DDPM}}$ and $t_{100\text{-DDPM}}$ to predict activation masks. When $t_{250\text{-DDPM}} = t_{100\text{-DDPM}}$, the denoising process is at the same stage, resulting in identical activation masks from TWD.

**Question3: Are there any suggestions about the selection of $\lambda$?**

*a)* Depending on computational resources, users may select different $\lambda$ values during fine-tuning to balance efficiency and performance.

*b)* We recommend initially setting $\lambda = 0.7$, as it generally delivers comparable performance. If the results are satisfactory, consider reducing $\lambda$ (e.g., to 0.5) for further optimization. Conversely, if performance is inadequate, increasing $\lambda$ may be beneficial.

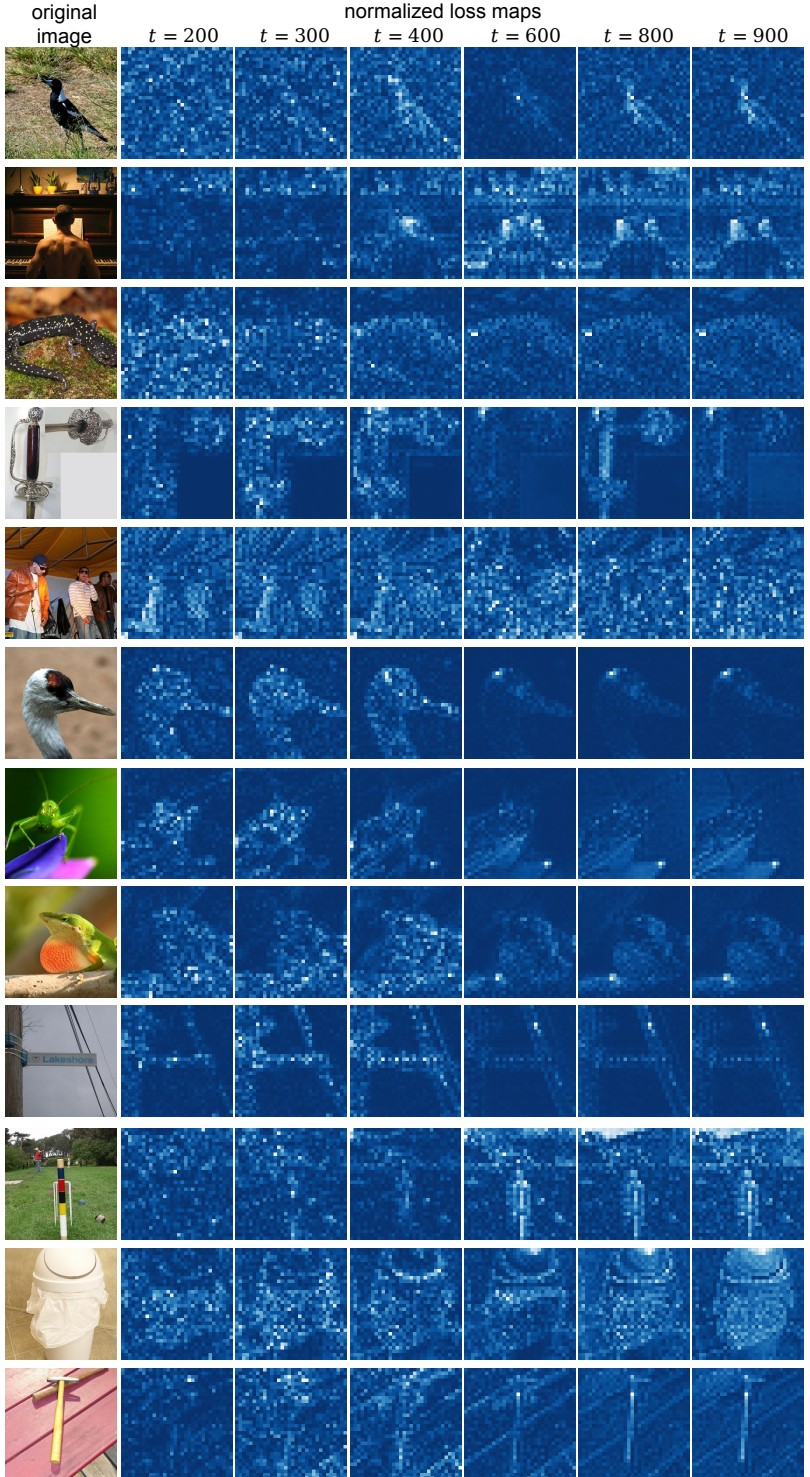

Figure 10: **Additional visualization of loss maps from DiT-XL.** The loss values are normalized to the range [0, 1]. Different image patches exhibit varying levels of prediction difficulty.

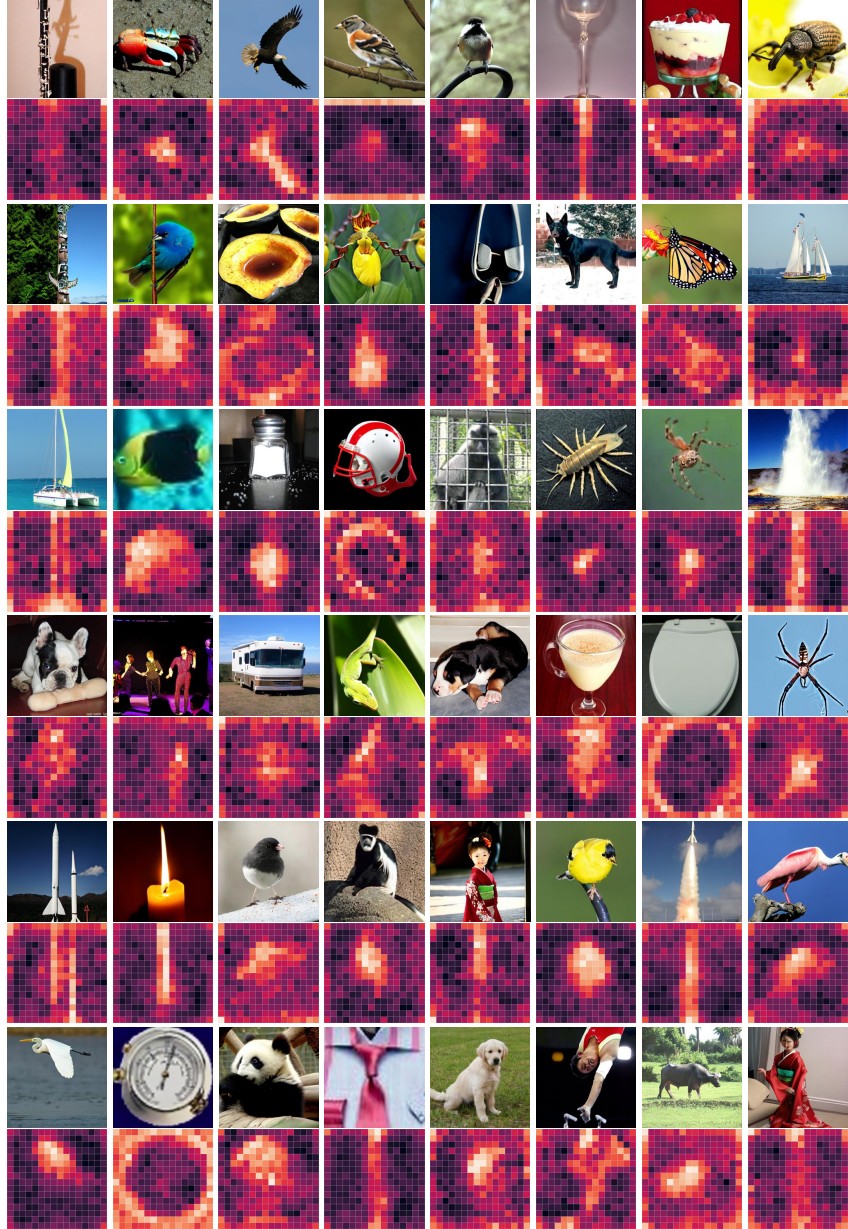

Figure 11: **Additional visualizations of computational cost across different image patches.** Complementary to Figure 6, we visualize more generated images and their corresponding FLOPs cost across different image patches. The map is normalized to [0, 1] for clarity.

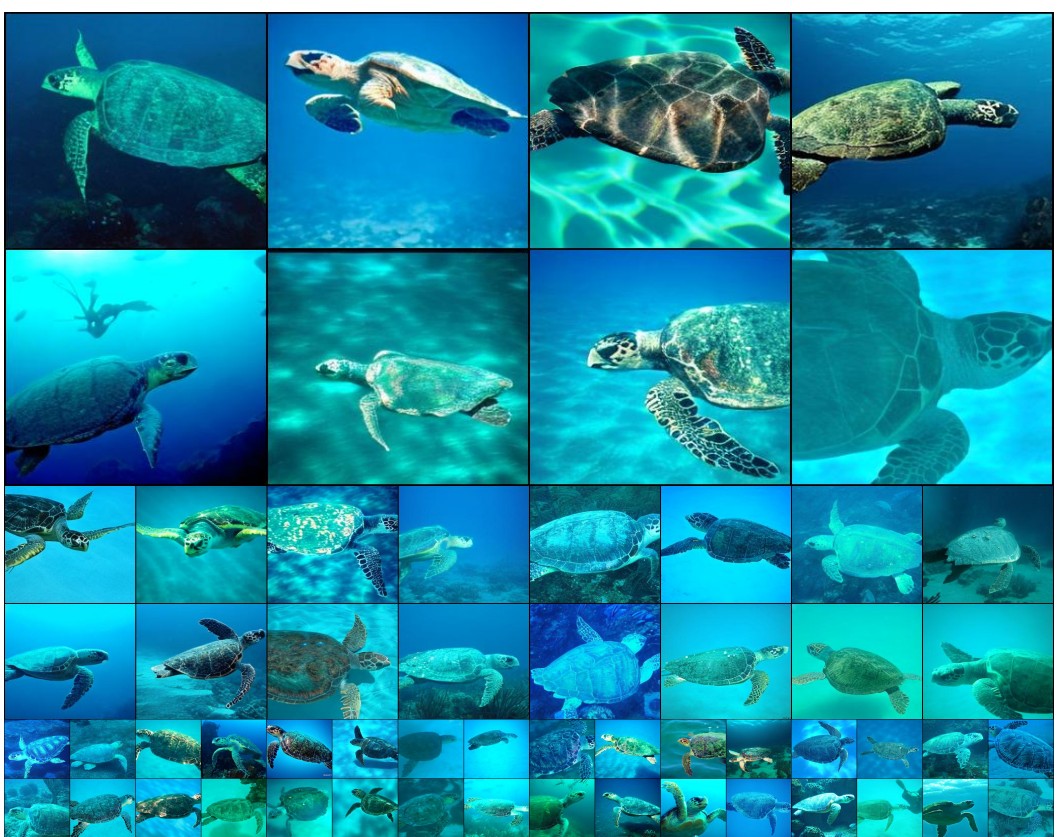

Figure 12: **Uncurated 256×256 DyDiT-XL$_{\lambda=0.5}$ samples. Loggerhead turtle (33).**

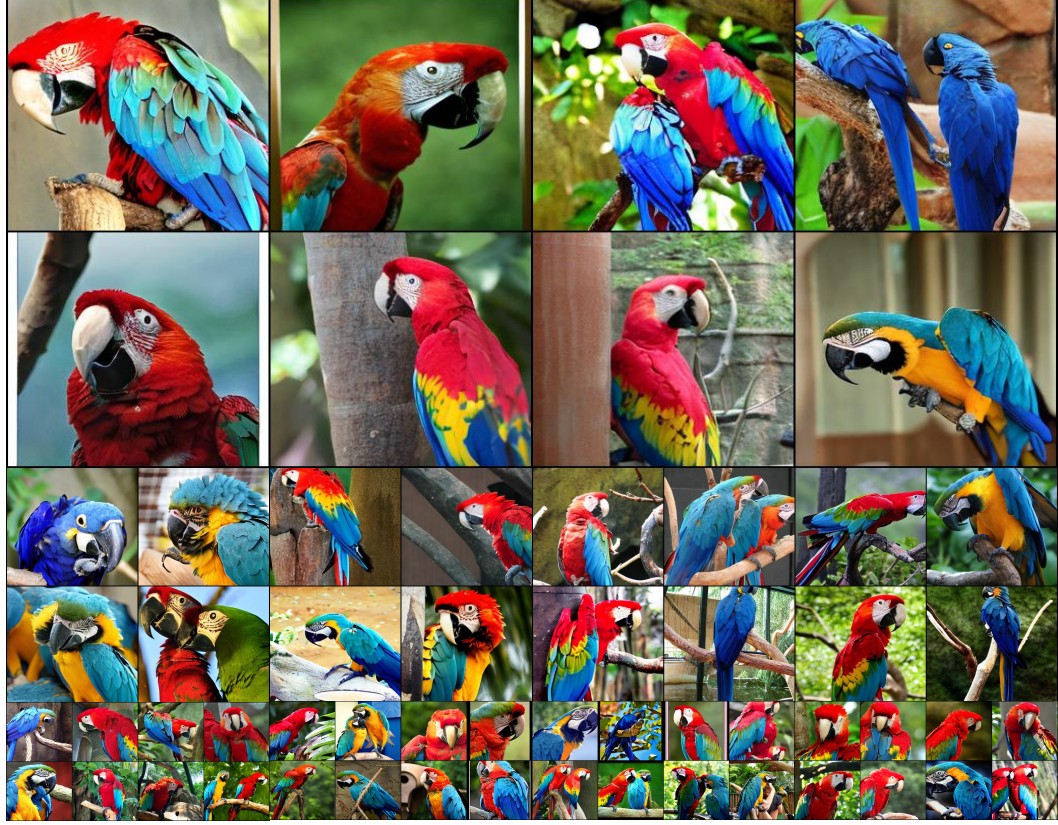

Figure 13: **Uncurated 256×256 DyDiT-XL$_{\lambda=0.5}$ samples. Macaw (88).**

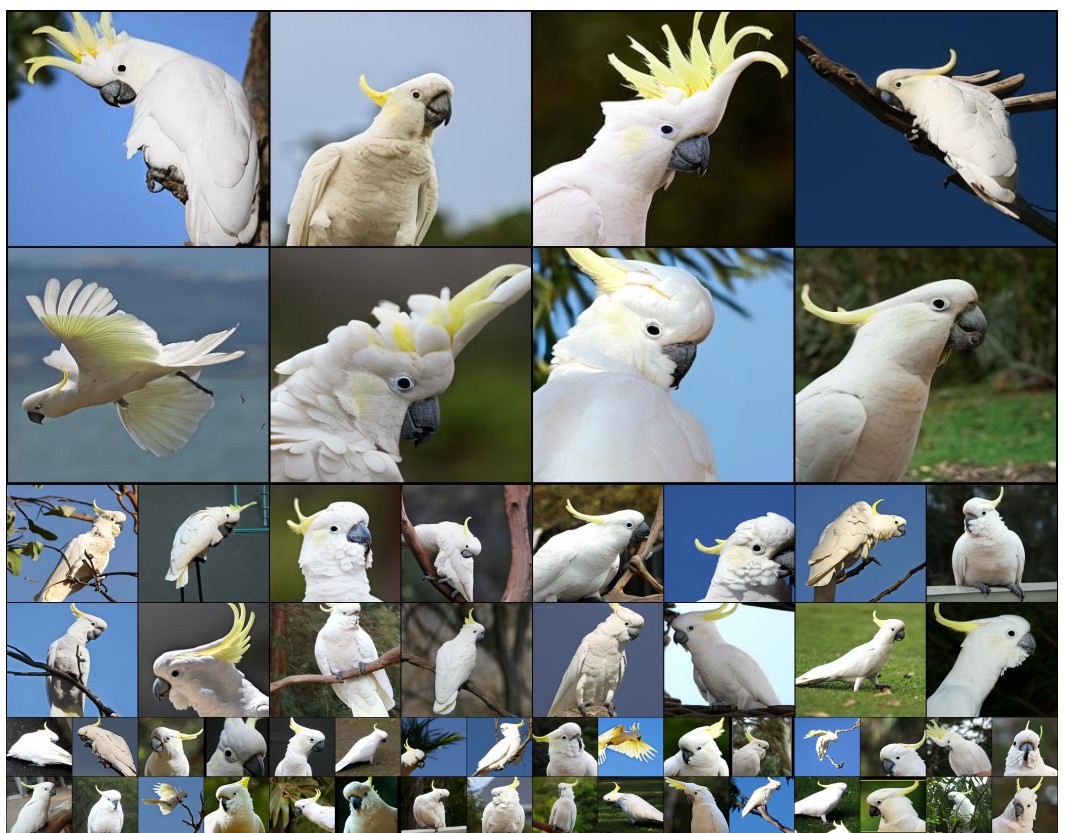

Figure 14: **Uncurated 256×256 DyDiT-XL$_{\lambda=0.5}$ samples. Kakatoe galerita (89).**

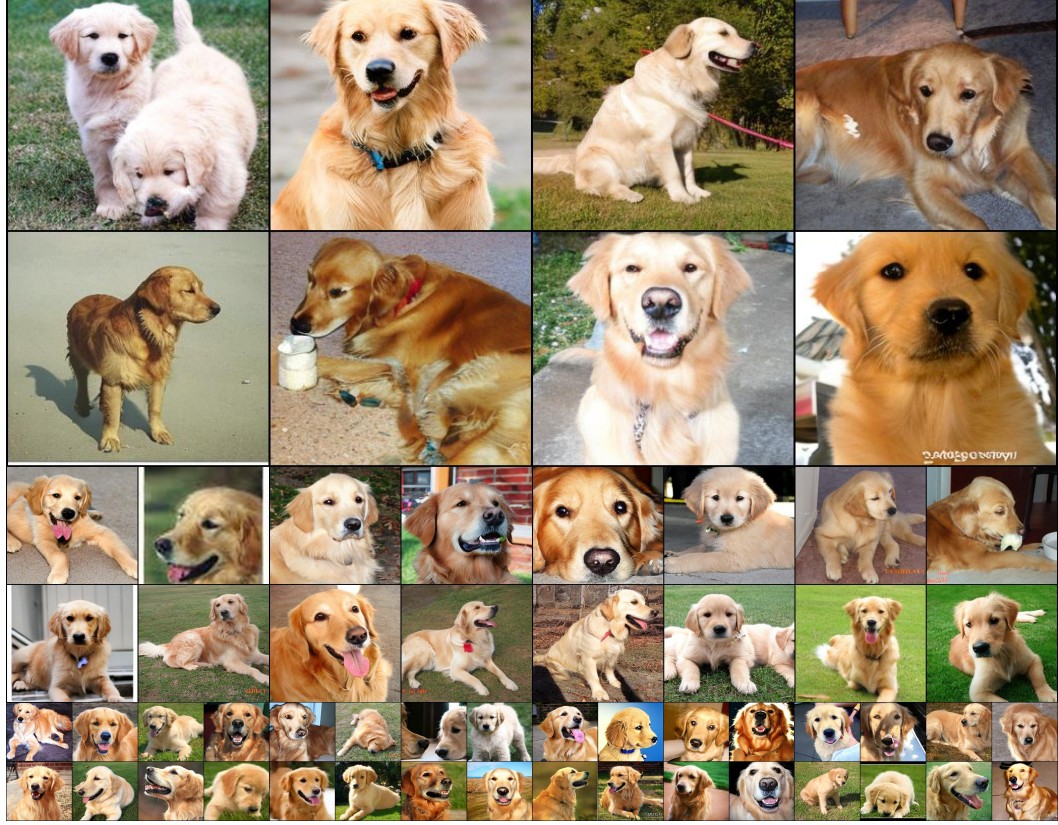

Figure 15: **Uncurated 256×256 DyDiT-XL$_{\lambda=0.5}$ samples. Golden retriever (207).**

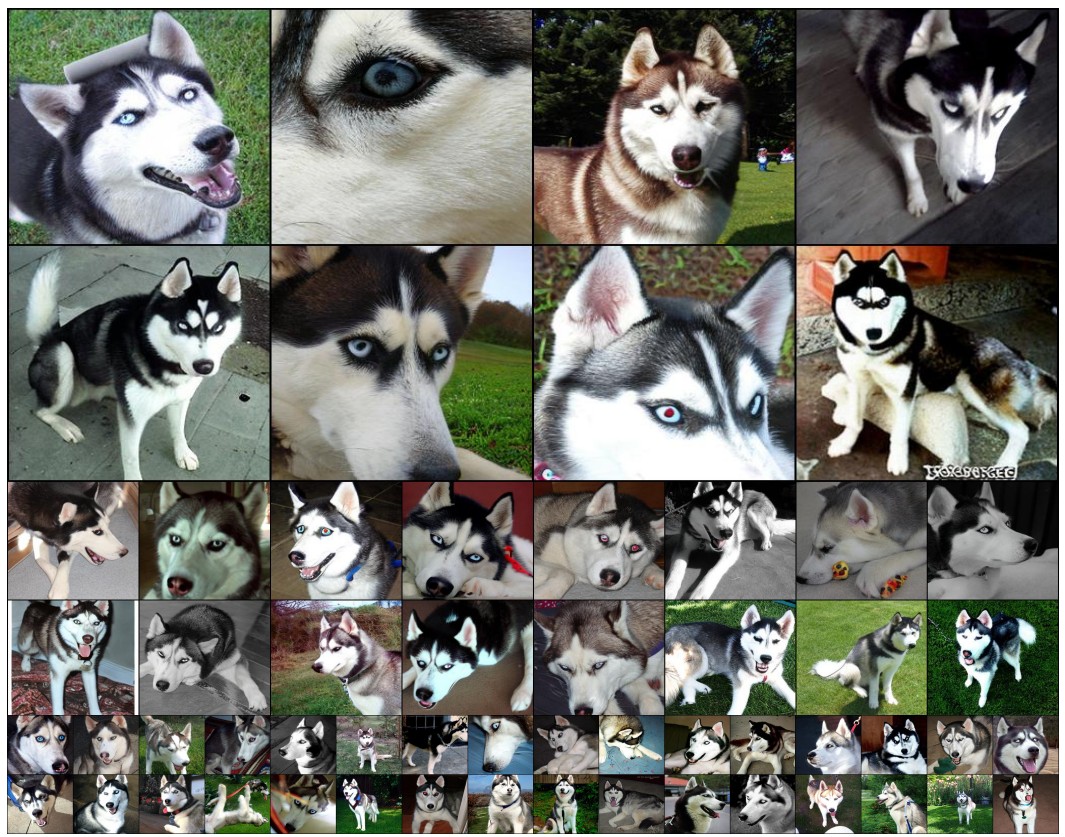

Figure 16: **Uncurated 256×256 DyDiT-XL$_{\lambda=0.5}$ samples. Siberian husky (250).**

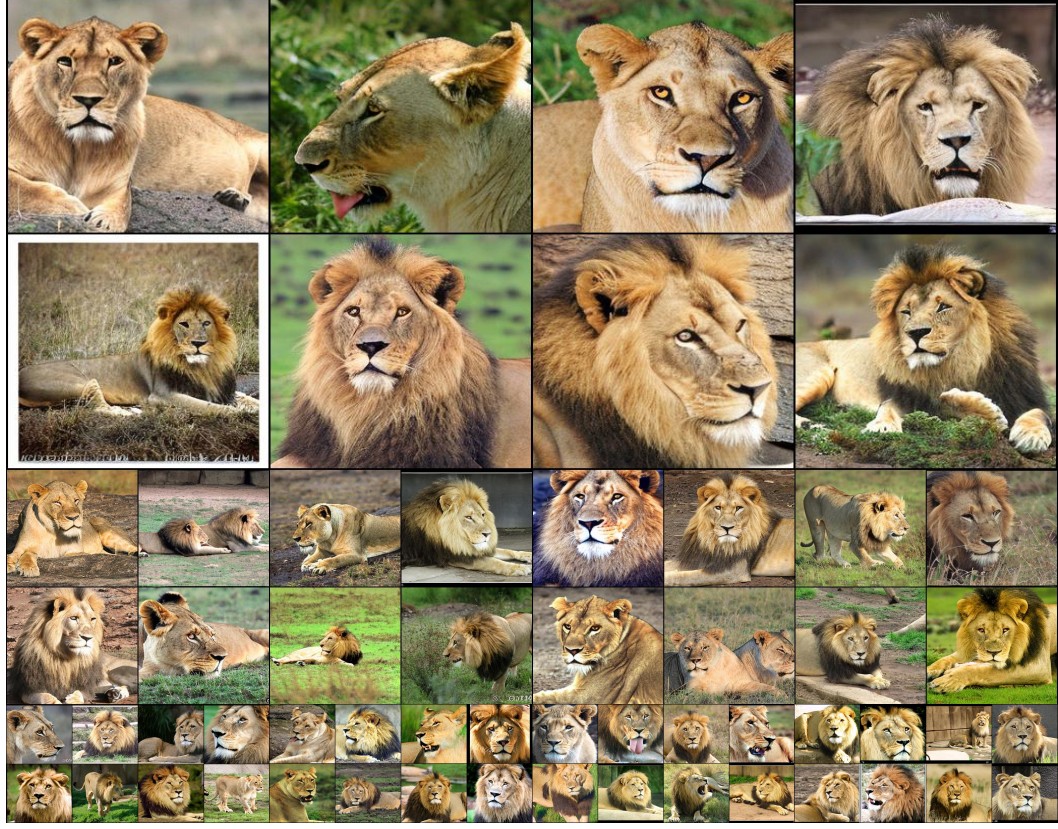

Figure 17: **Uncurated 256×256 DyDiT-XL$_{\lambda=0.5}$ samples. Lion (291).**

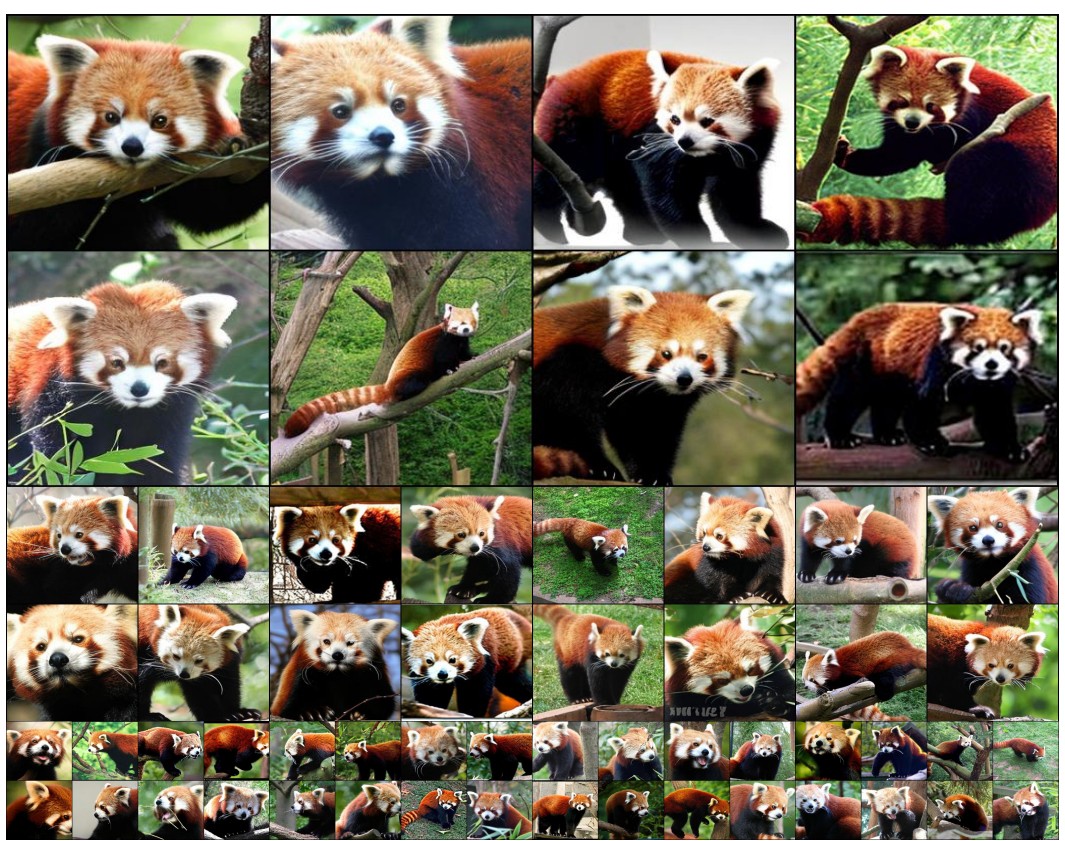

Figure 18: **Uncurated 256×256 DyDiT-XL$_{\lambda=0.5}$ samples. Lesser panda(387).**

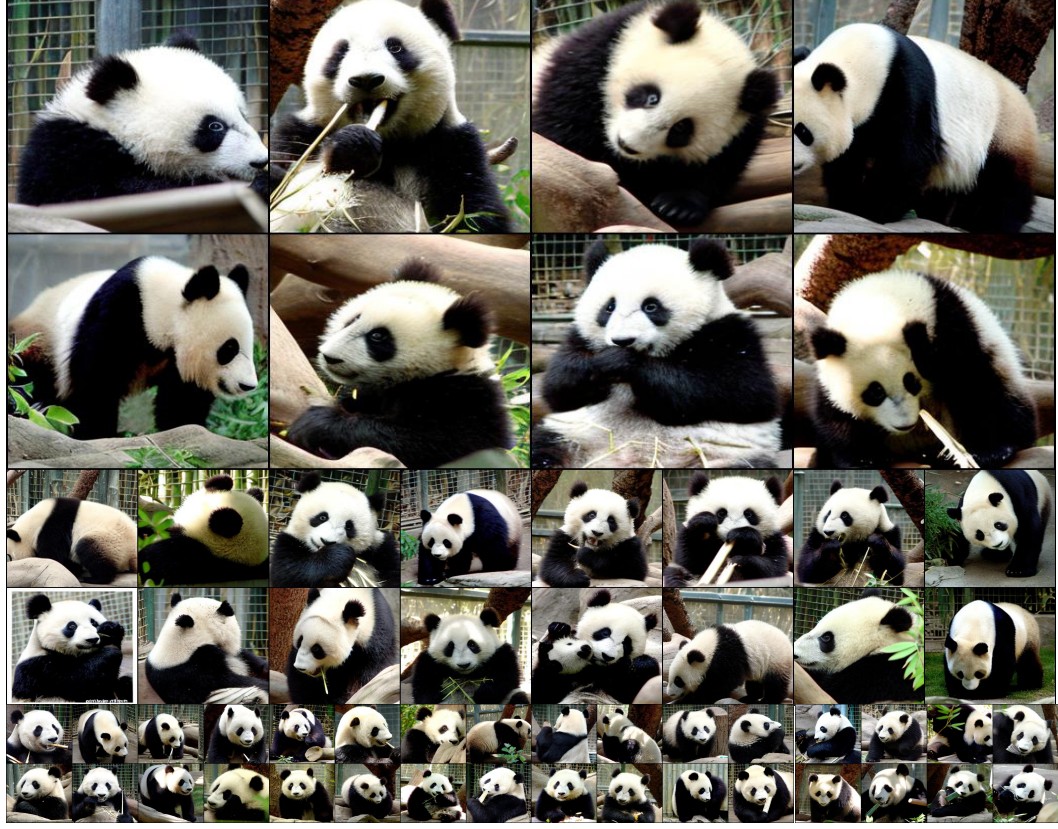

Figure 19: **Uncurated 256×256 DyDiT-XL$_{\lambda=0.5}$ samples. Panda (388).**

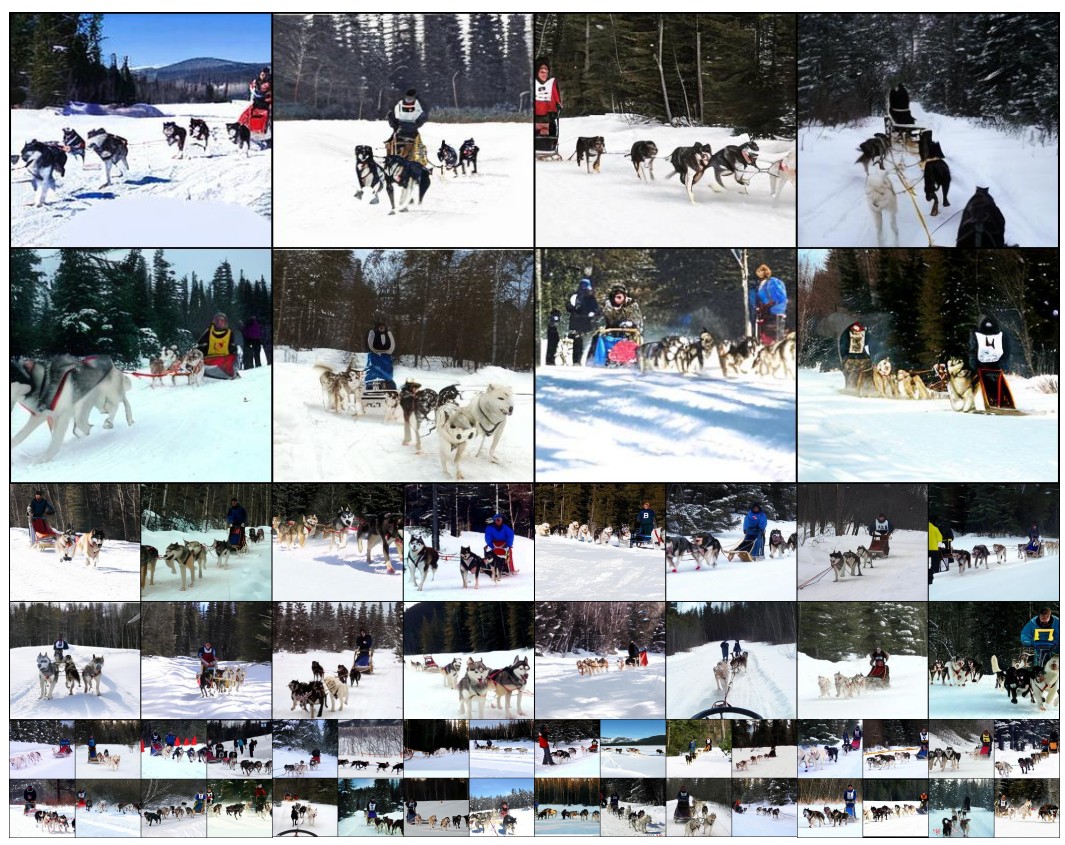

Figure 20: **Uncurated 256×256 DyDiT-XL$_{\lambda=0.5}$ samples. Dogsled (537).**

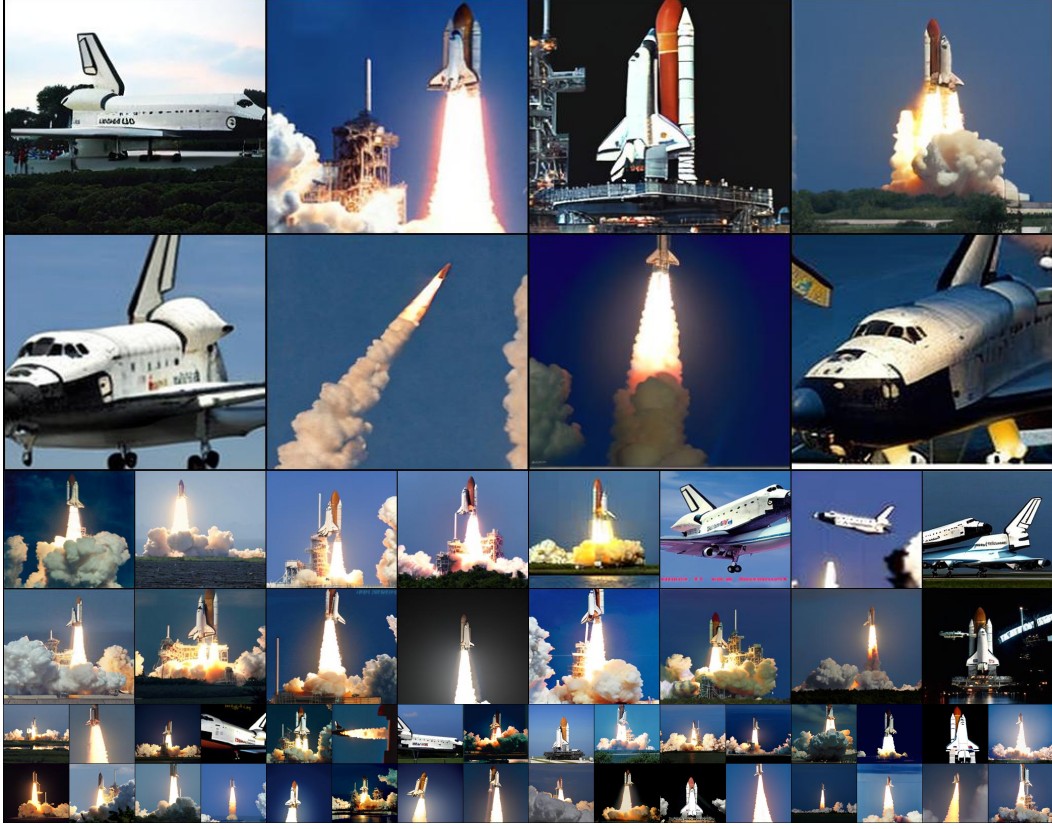

Figure 21: **Uncurated 256×256 DyDiT-XL$_{\lambda=0.5}$ samples. Space shuttle (812).**

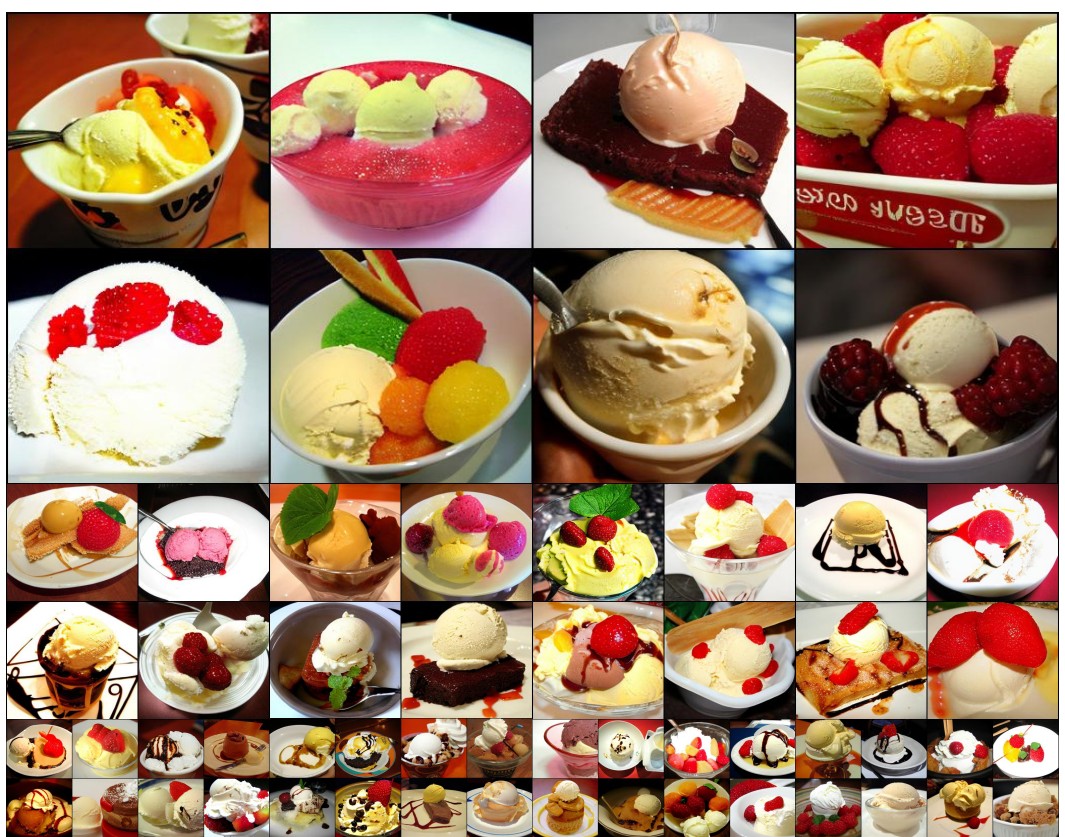

Figure 22: **Uncurated 256×256 DyDiT-XL$_{\lambda=0.5}$ samples. Ice cream (928).**

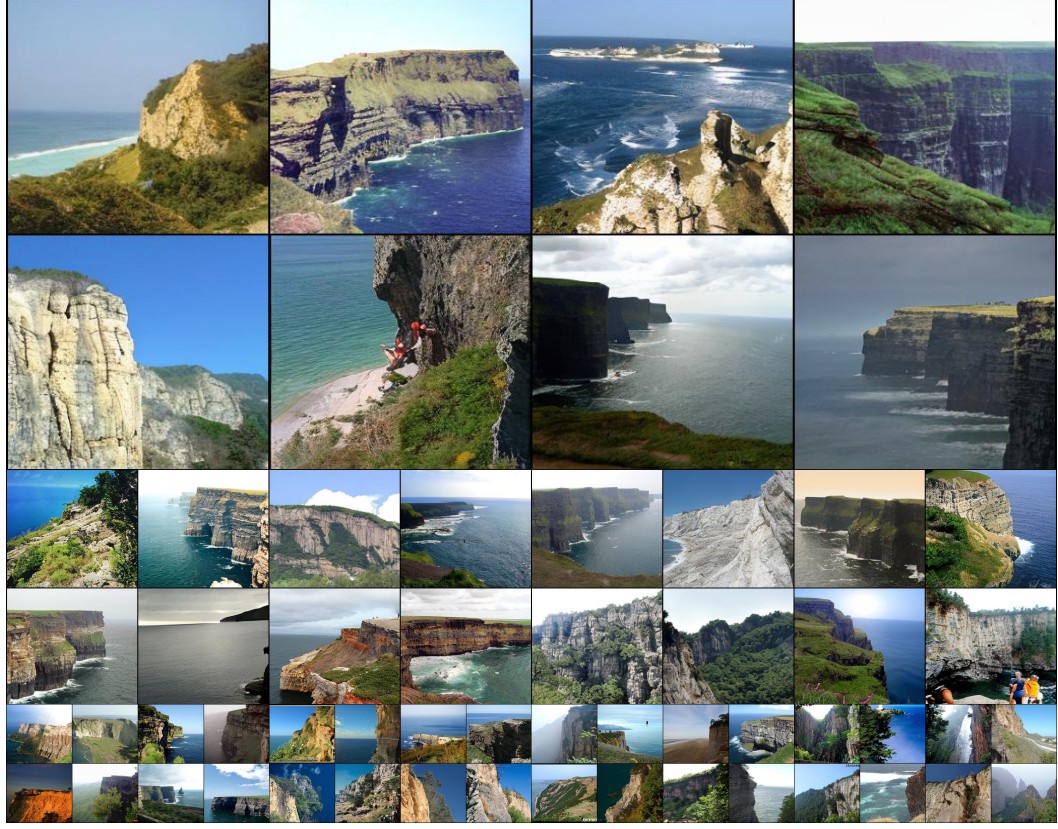

Figure 23: **Uncurated 256×256 DyDiT-XL$_{\lambda=0.5}$ samples. liff(972).**

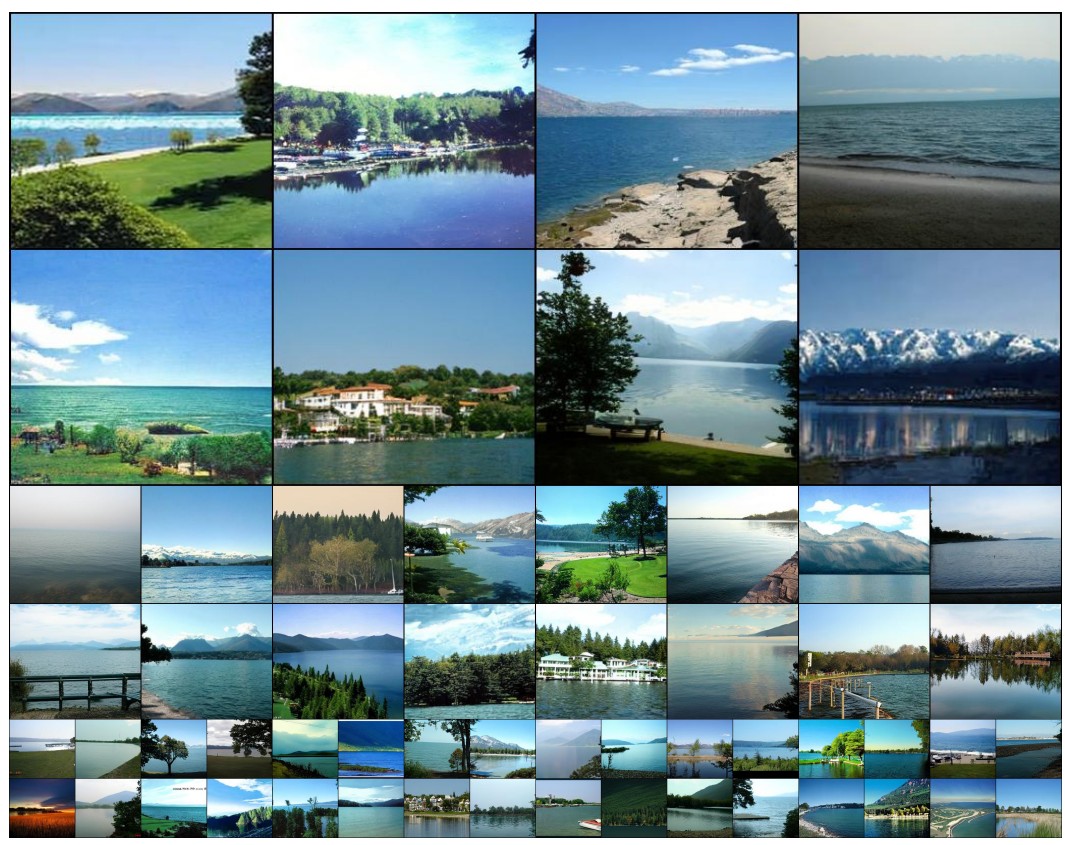

Figure 24: **Uncurated 256×256 DyDiT-XL$_{\lambda=0.5}$ samples. Lakeside (975).**

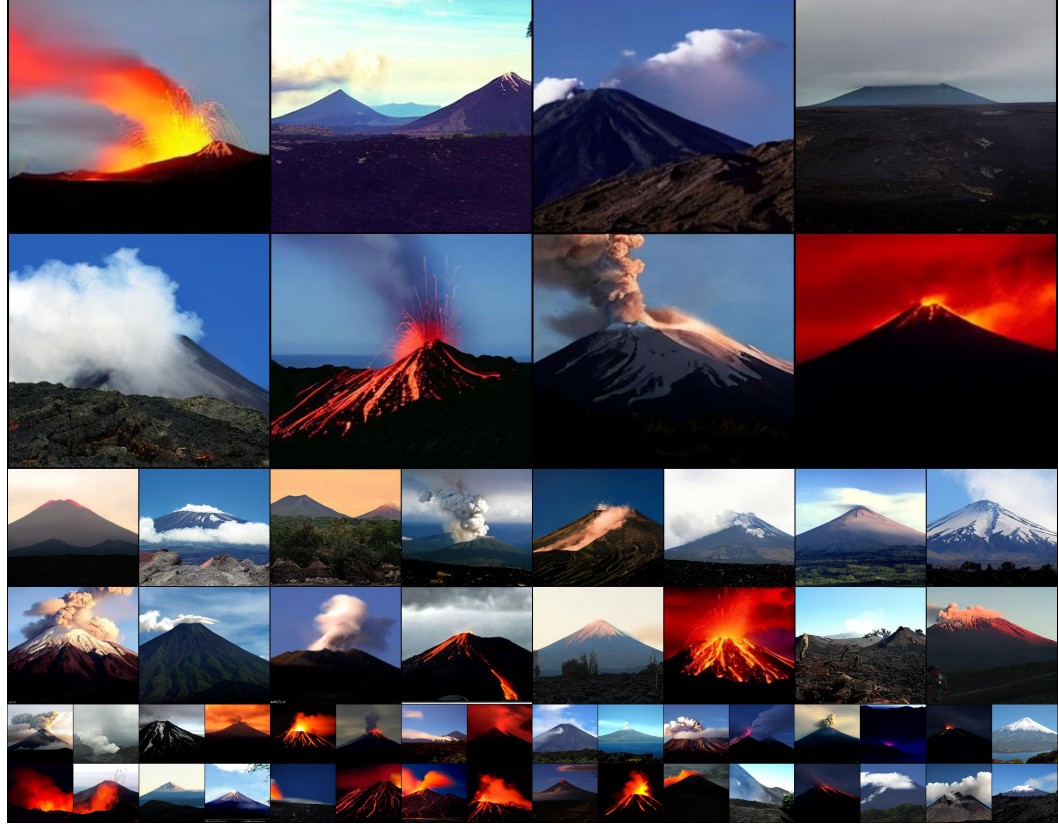

Figure 25: **Uncurated 256×256 DyDiT-XL$_{\lambda=0.5}$ samples. Volcano (980).**

