# OpenReview forum: "Dynamic Diffusion Transformer"
_ICLR.cc/2025/Conference — ICLR 2025 Poster_

### Official Review · Reviewer_TEwe · 2024-10-27

**Soundness:** 2
**Presentation:** 3
**Contribution:** 2
**Rating:** 3
**Confidence:** 5

**Summary:**

The authors propose Dynamic Diffusion Transformer (DyDiT) that dynamically adjusts its computation along both timestep and spatial dimensions during generation to reduce computational redundancy. This method contains two key components including a Timestep-wise Dynamic Width (TDW) approach and a Spatial-wise Dynamic Token (SDT) strategy. TDW is motivated by the observation in the difference of loss curve between DiT-S and DiT-XL across various time step. SDT is inspired by loss map showing patch-wise difficulty of noise prediction. Extensive experiments are conducted to show the effectiveness of the proposed method. But I fail to see the code or pseudocode.

**Strengths:**

1. The paper is easy to follow and well-written with many figure to clearly illustrate the idea of the paper as well as the results.

2. The observation is interesting and the method is intuitive and effective.

3. The authors conduct extensive experiments on various datasets, such as ImageNet and Food dataset, and DiT variants like DiT-S and DiT-XL.

**Weaknesses:**

1. This method is built on a well-pretrained DiT model, which may restrict its application. Is this method applicable for training a DiT model from scratch?

2. In Fig 3 and Appendix A 3, the comparison with other pruning methods may not be convincing. The authors is recommended to compare the proposed method with other dynamic methods such as repurposed ITOP [1], SViTE, and S2ViTE [2]. And why the authors do not provide the results of lambda over 0.7 in DiT-XL.

3. It seems that the authors do not choose a consistent lambda value for studies, such as 0.4 in Table 9, 0.5 in Table 10, and 0.9 in Table 14. In other words, this hyperparameter is relatively sensitive to obtain satisfying results under different settings. It would be better to provide some suggestions about the selection of this hyperparameter.

4. The comparison of maintaining the same training iterations for models in Appendix B3 may be partially reasonable. In addition to comparing DyDiT with DiT, the authors should also compare DyDiT with a DiT variant that incorporates the same gating module as DyDiT in the same position, multiplies the module output with the original input, and then uses this weighted input as the final input. In this way, we may keep the same number of parameters and further show the effectiveness of the proposed method.

5. Though the authors combine DyDiT with LCM (a 4-step model) to show the effectiveness of proposed method, inevitably LCM is contradictory with the motivation of DyDiT because DyDiT, especially the TDW, is inspired by the observation in a 250-step model. I speculate SDT may play a dominant role. In a nutsheel, this method may have a weak influence to those 1-step, 2-step, or 4-step models like LCM.

[1] Do We Actually Need Dense Over-Parameterization? In-Time Over-Parameterization in Sparse Training.

[2] Chasing Sparsity in Vision Transformers: An End-to-End Exploration.

**Questions:**

1. Why there is a significant performance drop when using SDT only (II). Could the authors provide a detailed explanation?

2. How about using layer-skip only in Table 3 III (layer-skip), i.e., removing TDW?

3. Could the authors provide a detailed description about how to obtain the results in Fig 6.

4. Why the authors use 100 DDPM in Table 12 instead of 250 like DiT. How about the results using 250 DDPM?

5. For other questions, please see weakness.

Minor issue: Line954 our-> Our

---

> ### Author Response · Authors · 2024-11-20
> **Response to Reviewer TEwe (1/4)**
>
> > W1: This method is built on a well-pretrained DiT model, which may restrict its application. Is this method applicable for training a DiT model from scratch?
>
>
> We appreciate this insightful question. As a dynamic neural network, our method can be trained from scratch. Nevertheless, **building upon a well-pretrained model would present several advantages**. The reasons are outlined below:
>
>
> **Reasons**:
>
> + **Building from a pre-trained model reduces training cost**. The DiT-XL model is **pre-trained over 7 million iterations** (Lines 888-897) with a batch size of 256. Our model can be adapted with **less than 3% additional fine-tuning iterations**, saving approximately 6.79 million iterations. This approach is **particularly beneficial when computational resources are limited.**
>
> + **Fine-tuning is a common and effective strategy**. Since the cost of **training a diffusion model from scratch is high and unaffordablee** for most users, fine-tuning from a pre-trained model is well adopted in various applications [1, 2, 3, 4]. Along with them, our method is also built based on pre-trained model to reduce the training cost.
>
>
>
>
> We sincerely appreciate the suggestion to train from scratch. In this paper, we primarily focus on **exploring the potential of dynamic neural networks within diffusion transformer (DiT)**. Employing a pre-trained model **facilitates this exploration**. We plan to allocate additional computational resources to investigate the effectiveness of training a DyDiT from scratch in future work.
>
>
>
>
>
> **Reference**:
>
> [1] Adding Conditional Control to Text-to-Image Diffusion Models, ICCV 2023.
>
> [2] Difffit: Unlocking transferability of large diffusion models via simple parameter-efficient fine-tuning, ICCV 2023.
>
> [3] IP-Adapter: Text Compatible Image Prompt Adapter for Text-to-Image Diffusion Models, 2023.
>
> [4] Tune-a-video: One-shot tuning of image diffusion models for text-to-video generation, 2023
>
>
> > W2: In Fig 3 and Appendix A 3, the comparison with other pruning methods may not be convincing. The authors is recommended to compare the proposed method with other dynamic methods such as repurposed ITOP, SViTE, and S2ViTE.
>
>
> Thanks. The comparison with pruning methods in Fig3 and Appendix A3 are inspired by [1], which also include these pruning methods in its experiments. Additionally, we would like to clarify that the term "dynamic" in methods ITOP, SViTE, and S2ViTE  refer to **dynamic sparse training**, which **fundamentally differs** from the **dynamic architecture** in DyDiT. The distinctions are outlined below:
>
>
> | model | improve inference efficiency | obtained architecture| target model| target task
> |-|-|-|-|-
> | ITOP,  SViTE and S2ViTE  | ✅| static: data-independent|ResNet, ViT| image preception e.g. classification
> | DyDiT (ours) |✅| dynamic: timestep-dependent(TDW) and data-dependent (SDT)|DiT| image diffusion generation
>
> **Analysis:**
> + Starting from a sparse network, ITOP, SViTE, and S2ViTE explore parameter connectivity during training, resulting in a **static sparse network as the final output**. Thus, networks obtained from these methods are **static**, unlike the dynamic architecture of the proposed DyDiT.
>
> + ITOP is tailored for ResNet50, while SViTE and S2ViTE are specifically designed for the transformer-based ViT model, focusing on unstructured and structured sparsity, respectively. Consequently, they are primarily **intended for image perception tasks**, rather than image diffusion generation as in DiT.
>
>
>
>
> Despite these inherent differences, we conducted an experiment by **applying S2ViTE to DiT for more comprehensive comparison**. We chose S2ViTE due to its hardware-friendly structured sparsity, which is relevant to the dynamic head and channel groups in DyDiT.
>
> **Settings:**
> We fine-tune "DiT pruned w/ S2ViTE" by 100,000 iteractions,  twice the duration used for DyDiT-S $_{\lambda=0.5}$, since the original S2ViTE suggests that extended training enhances its dynamic sparse training. Other settings follows Section4.2.
>
>
> | model                | FID  ↓ | FLOPs (G) ↓|
> | -------------------- | ----- | --------- |
> | DiT-S                | 21.46 | 6.07      |
> | DiT pruned w/ S2ViTE |  48.90    |    3.05        |
> | DyDiT-S $_{\lambda=0.5}$ |  28.75  |      3.16   |
>
> **New**: Results of incorporating DiT-B/XL and S2ViTE are uploaded. Please refer to the comments titled 'Results of Incorporating DiT-B and S2ViTE' and 'Results of Incorporating DiT-XL and S2ViTE' below.
>
>
> **Conclusion**:
>
> Resutls reveal that the proposed "DyDiT-S $_{\lambda=0.5}$" can surpass "DiT pruned w/ S2ViTE" obviously.  This demonstrate the **superiority of our dynamic architecture over static architecture**.
> Due to time constraints, we have only acquired results for DiT-S. The training process for applying SViTE to DiT-B and DiT-XL is ongoing. **We will provide updated results promptly.**
>
>
> **Reference**:
>
> [1] Structural pruning for diffusion models, NeurIPS 2023

---

> ### Author Response · Authors · 2024-11-20
> **Response to Reviewer TEwe (2/4)**
>
> >W3: Why the authors do not provide the results of lambda over 0.7 in DiT-XL.
>
>
> Thanks. The $\lambda$ controls the FLOPs of DyDiT. In DiT-XL, results in Figure3 \(c\) show that DyDiT **$\lambda=[0.5, 0.6, 0.7]$ can achieve performance comparable to the original DiT**. Further increasing the $\lambda$ will result in additional FLOPs and **hurt the FLOPs-performance trade-off**. Based on the suggestion, we further conduct experiments with $\lambda=0.8$ and $\lambda=0.9$.
>
> **Settings**:
> We follow the setting in Appendix B1 to conduct experiments on DyDiT $_{\lambda=0.8}$ and DyDiT$ _{\lambda=0.9}$.
>
>
>
> | model | s/image |  acceleration |   FLOPs (G) ↓ |FID↓| FID ∆ ↓  |
> | -------- | -------- | -------- | --- | --- |-- |
> | DiT-XL     |10.22| 1.00 × |118.69| 2.27| +0.00
> |DyDiT-XL$_{\lambda=0.9}$|9.73| 1.05 ×|110.73|2.15|-0.12
> |DyDiT-XL$_{\lambda=0.8}$|8.66| 1.18 ×|96.04|2.13|-0.14
> |DyDiT-XL$_{\lambda=0.7}$| 7.76| 1.32 ×| 84.33| 2.12| -0.15
> |DyDiT-XL$_{\lambda=0.6}$| 6.86| 1.49 ×| 67.83| 2.18| -0.09
> |DyDiT-XL$_{\lambda=0.5}$| 5.91| 1.73 ×| 57.88| 2.07| -0.20
> |DyDiT-XL$_{\lambda=0.3}$| 4.26| 2.40 ×| 38.85| 3.36| +1.09
>
> **Analysis**:
> + We find that DyDiT with $\lambda=0.8$ and $\lambda=0.9$ also achieves **performance comparable to the original DiT**, but **incurs additional FLOPs** compared to settings like $\lambda=0.7$.
> + Setting $\lambda$ between 0.5 and 0.7 offers a **better trade-off** between computational cost and performance.
>
>
> These results are included in **Appendix B1**.
>
>
>
>
>
>
>
>
> > W4: It seems that the authors do not choose a consistent lambda value for studies, such as 0.4 in Table 9, 0.5 in Table 10, and 0.9 in Table 14. In other words, this hyperparameter is relatively sensitive to obtain satisfying results under different settings.
>
> Thanks. We would like to clarify the impact of $\lambda$. Since $\lambda$ **controls the FLOPs** of our method (via $\mathcal{L}_{\text{FLOPs}}$ in Equation 6 of the main paper), it does influence performance. Allocating **more FLOPs** (a larger $\lambda$) generally **enhances generation performance**, while **fewer FLOPs** (a smaller $\lambda$) can result in **relatively lower performance**.
>
>
>
>
> Since **tasks vary in difficulty** and **models differ in computational redundancy levels**, we employ different $\lambda$ values to **ensure DyDiT achieves performance comparable to its counterparts** in these studies. We list the explaintation to each question in detail.
>
> **Explanation**:
>
> |question|explaintation|
> |-|-
> | 0.4 in Table 9| In Table 9, since **generation on CIFAR-10 is relatively easy**, there is more redundancy in the original U-ViT-S/2. Our method with $\lambda=0.4$ performs comparably to the original U-ViT-S/2 and surpasses other pruning methods. |
> |0.5 in Table 10|In Table 10, since **generation on ImageNet is more challenging** than CIFAR-10, the redundancy in U-ViT-H/2 is also less. In this scenario, we set $\lambda$ to 0.5 for our method to achieve comparable performance.|
> |0.9 in Table 14| Since the **LCM performs generation with only 4 steps**, further **accelerating it without impacting performance is more challenging** than other tasks and models. We experimented with different $\lambda$ values and found that $\lambda=0.9$ can accelerate the LCM without sacrificing performance.|
>
>
>
>
>
> > W5: It would be better to provide some suggestions about the selection of this hyperparameter.
>
>
> Thanks. We present our suggestions below:
>
> + Depending on computational resources, users may **select different $\lambda$** values during fine-tuning to **balance efficiency and performance**.
> + We recommend **initially setting $\lambda = 0.7$**, as it generally delivers comparable performance. If the results are satisfactory, consider reducing $\lambda$ (e.g., to 0.5) for further optimization. Conversely, if performance is inadequate, increasing $\lambda$ may be beneficial.
>
>
> These recommendations are included into **“Frequent Questions” section in the Appendix**

---

> ### Author Response · Authors · 2024-11-20
> **Response to Reviewer TEwe (3/4)**
>
> >W6: The comparison of maintaining the same training iterations for models in Appendix B3 may be partially reasonable. In addition to comparing DyDiT with DiT, the authors should also compare DyDiT with a DiT variant that incorporates the same gating module as DyDiT in the same position, multiplies the module output with the original input, and then uses this weighted input as the final input.  In this way, we may keep the same number of parameters and further show the effectiveness of the proposed method.  In this way, we may keep the same number of parameters and further show the effectiveness of the proposed method.
>
>
> We appreciate this suggestion and incorporate the recomonded experiment to ensure a more fair and reasonable comparison.
>
> **Setting**
>
> We develop the model according to the reviewer's instructions:
>
> 1. We build a DiT-XL with the same router modules as DyDiT in the same position.
> 2. The output from the router modules are multiplied with the original input to serve as the final input of the DiT block.
> 4. This model comprises 678M parameters, identical to DyDiT, with an additional 3M parameters compared to the original DiT.
> 5. We perform fine-tuning by 150000 iterations for this model, denoted as DiT-XL$^\prime$.
>
>
> |model| pre-trained iterations| fine-tuning iterations| FLOPs (G) ↓| FID ↓| FID ∆ ↓
> |-|-|-|-|-|-|
> |DiT-XL| 7,000,000| -| 118.69| 2.27| +0.00
> |DiT-XL| 7,000,000| 150,000 (2.14%)| 118.69| 2.16| -0.11
> |DiT-XL$^\prime$|7,000,000|150,000 (2.14%)|118.69|2.15|-0.12
> |DyDiT-XL$_{\lambda=0.7}$| 7,000,000| 150,000 (2.14%)| 84.33| 2.12| -0.15
>
>
> **Conclusion:**
>
> This model also achieves performance comparable to the original DiT with additional 150,000 fine-tuning iterations. However, compared with the proposed DyDiT, it **can not reduce the computational cost during inference**.
>
>
> This experiment is included in **Appendix B3**.
>
>
>
>
>
> >W7: Though the authors combine DyDiT with LCM (a 4-step model) to show the effectiveness of proposed method, inevitably LCM is contradictory with the motivation of DyDiT because DyDiT, especially the TDW, is inspired by the observation in a 250-step model. I speculate SDT may play a dominant role. In a nutsheel, this method may have a weak influence to those 1-step, 2-step, or 4-step models like LCM.
>
>
>
> We sincerely appreciate this insightful question. We present our explanations below:
>
>
> + **Motivation of combining our method with LCM**: We acknowledge that our method, incorporating TDW and SDT, is derived from observations of DiT with DDPM, which is different from LCM. Nevertheless, we notice that LCM and our method accelerate the diffusion model **from different perspectives**. Specifically, LCM focuses on reducing the number of sampling steps, whereas our method aims to reducing the average FLOPs per step. Consequently, these methods **may be compatible to some extent**, prompting us to include experiments on LCM in the Appendix.
>
>
> + **TDW's role in 4-step LCM**:
> Inspired from the question, we record the activation of TDW in "DyDiT-XL$_{\lambda=0.9}$ + 4-step LCM" during generation. We also report the FLOPs allocated at each timestep.
> 	|          | $t$=3 | $t$=2 | $t$=1 | $t$=0 |
> 	| -------- | ----- | ----- | ----- | ----- |
> 	| TDW-attn      | 100%| 100% | 100% | 100% |
> 	| TDW-mlp      | 95.98% | 100% | 99.33% | 96.21% |
> 	| FLOPs(G) | 100.989 | 106.664 | 106.270 | 103.816 |
> 	+ We can observe that **channel groups are activated differently at various timesteps**, while attention block heads remain consistently activated. This shows **TDW's ability to adjust model width dynamically for different timesteps in LCM.**
> 	+ Combining TDW and SDT, **the FLOPs varies across timesteps**. Notably, at $t=3$ and $t=0$, fewer FLOPs are used, indicating dynamic resource allocation of our method.
>
> + **Further reducing LCM steps is challenging:** We attempt to reduce the sampling steps in the LCM from 4 to 3, 2, and even 1 step. Nevertheless, our observations indicate a noticeable decline in the visual quality of images generated by both DyDiT and DiT models. This demonstrates that **further reduction in sampling steps of LCM poses significant challenges not only for DyDiT and its TWD but also for the original DiT**.
>
> + **Exploration of incorporating dynamic neural network with LCM**: In Lines 1069-1071, we acknowledge that DyDiT’s training relies on **noise prediction at different timesteps**, which is absent in LCM. Including the results of DyDiT+LCM in the Appendix aims to **preliminarily investigate the potential of dynamic neural networks in distillation-based samplers**. The preliminary findings suggest that integrating the dynamic architecture with distillation-based samplers such as LCM, could be a **promising future direction**.
>
> **Conclusion**:
>
> Integrating our method with LCM is not the primary focus of this paper. However, it suggests a promising direction to further accelerate DiT. Therefore, we include this experiment in the appendix to encourage future research.

---

> ### Author Response · Authors · 2024-11-20
> **Response to Reviewer TEwe (4/4)**
>
> > Q1: Why there is a significant performance drop when using SDT only (II). Could the authors provide a detailed explanation?
>
> Thanks. The reduction in FLOPs in "II" can **only be achieved through token bypass in MLP blocks**. With the target FLOPs ratio λ=0.5, a majority of tokens in "II" **must bypass the MLP blocks to satisfy the FLOPs constraint**. Consequently, the MHSA blocks are left to handle most tokens, which substantially impacts performance [1] (Lines 431-460). We use \emph{} to highlight this explanation in our revision.
>
>
> **Reference**:
>
> [1] Attention is not all you need: Pure attention loses rank doubly exponentially with depth, 2021
>
>
>
>
>
>
>
>
> > Q2: How about using layer-skip only in Table 3 III (layer-skip), i.e., removing TDW?
>
> Thanks for this valuable question. Based on this, we conduct an experiment by removing TDW from III (layer-skip), resuling in only layer-skip "IV".
>
>
> **Settings**:
> Following the setting in Table3, we conduct experiments on DyDiT-S$_{λ=0.5}$. All models evoke around 3.16 GFLOPs.
>
> | model  | TWD  | SDT|layer skip|ImageNet|Food|Artbench|Car|Bird|#Average
> |--------|--|---|-------|-------|-------|-|-|-|-
> |II|❌|✅|❌|70.06| 23.79| 52.78| 16.90| 12.05| 35.12
> |III (layer-skip) | ✅| ❌|✅|30.95|17.75| 23.15| 10.53| 9.01| 18.29
> |IV  | ❌| ❌|✅|77.23|27.21|40.31|41.43|23.65|41.96
>
>
> **Analysis**:
> + Comparing III (layer-skip) with IV demonstrates that removing the proposed TWD results in degraded performance. This is due to the **model needing to bypass numerous MLP blocks for all tokens** to meet the target FLOPs, which significantly impacts performance [1]. This explanation aligns with our discussion in Q1.
>
> + When comparing IV with II, it is evident that IV underperforms relative to II. This is because **different regions of an image encounter distinct challenges in noise prediction** (see Section 1 of the main paper). The uniform token processing strategy of **layer-skip in IV fails to accommodate this heterogeneity**, whereas the spatial-wise dynamic token (SDT) approach of II effectively addresses it.
>
>
>
>
>
>
> **Conclusion**:
> Layer-skipping is not an effective approach for enhancing the efficiency of DiT.
>
>
>
> **Reference**:
> [1] Attention is not all you need: Pure attention loses rank doubly exponentially with depth, 2021
>
>
>
>
> > Q3: Could the authors provide a detailed description about how to obtain the results in Fig 6.
>
> Thanks.
>
> + The FLOPs difference between tokens (patches) **results from varying costs in MLP blocks**. We accumulate the FLOPs for each token in each MLP block throughout the generation process, based on channel group activation in TWD and token activation in SDT.
> + We then **normalize these results to [0, 1]** and plot the heatmap.
>
>
>
>
>
>
>
>
>
>
>
>
>
> >Q4: Why the authors use 100 DDPM in Table 12 instead of 250 like DiT. How about the results using 250 DDPM?
>
> We appreciate these valuable questions. In Table 12, we used 100 DDPM timesteps to facilitate the experiments and save time. Based on the suggestion, we present the results using 250 DDPM timesteps below:
>
> **Setting**:
> Image generation at 512×512 resolution on ImageNet. Following DiT, We sample 50,000 images and leverage FID to measure the generation quality. FLOPs represent the computational cost per sampling step. For our method, we report the average FLOPs during generation.
>
> | model  | sampling steps|FLOPs(G)   | FID
> |--------|-----|-|-------
> |DiT  | 100 DDPM steps |514.80| 3.75
> |DyDiT $_{\lambda=0.7}$ (ours) | 100 DDPM steps |375.35| 3.61
> |DiT  | 250 DDPM steps |514.80| 3.04
> |DyDiT$_{\lambda=0.7}$ (ours) |  250 DDPM steps |375.05| 2.88
>
>
>
>
> **Analysis**:
> + We report the **average FLOPs per timestep**; thus, 100-DDPM results in **significantly fewer total FLOPs** than 250-DDPM.
> + The number of sampling steps introduces minor variations in the average FLOPs when comparing DyDiT with 100 DDPM steps to DyDiT with 250 DDPM steps.
> + With 250 DDPM timesteps, DyDiT also achieves performance comparable to the original DiT, demonstrating that our method is **robust across different numbers of timesteps**.
>
> Thanks for the suggestion from the reviewer,  the results of 250-DDPM are updated into **Table 12**.
>
>
>
>
> > Q5: Line954 our-> Our
>
> Thanks. This issue is revised. We will continue to carefully proofread the manuscript.

---

> ### Author Response · Authors · 2024-11-23
> **Results of incorporating DiT-B and S2ViTE**
>
> Dear reviewer TEwe,
>
> We would like to kindly remind that the results of incorporating DiT-B and S2ViTE are uploaded. Please refer to them for review.
>
>
> **Settings:**
> We fine-tune "DiT-B pruned w/ S2ViTE" by 200,000 iteractions,  twice the duration used for DyDiT-B $_{\lambda=0.5}$, since the original S2ViTE suggests that extended training enhances its dynamic sparse training. Other settings follows Section4.2.
>
> | model                | FID  ↓ | FLOPs (G) ↓|
> | -------------------- | ----- | --------- |
> | DiT-B                | 9.07  | 23.07      |
> | DiT-B pruned w/ S2ViTE |  19.84    |    11.54        |
> | DyDiT-B $_{\lambda=0.5}$ |  12.92 |       11.90   |
>
>
>
>
>
> Resutls reveal that the proposed "DyDiT-B $_{\lambda=0.5}$" can surpass "DiT-B pruned w/ S2ViTE" obviously.  This futher demonstrates the **superiority of our dynamic architecture over static architecture**.

---

> ### Author Response · Authors · 2024-11-25
> **Results of incorporating DiT-XL and S2ViTE**
>
> We fine-tune "DiT-XL pruned w/ S2ViTE" by 300,000 iteractions,  twice the duration used for DyDiT-XL $_{\lambda=0.5}$, since the original S2ViTE suggests that extended training enhances its dynamic sparse training. Other settings follows Section4.2.
>
> | model                | FID  ↓ | FLOPs (G) ↓|
> | -------------------- | ----- | --------- |
> | DiT-XL                | 2.27  | 118.69      |
> | DiT-XL pruned w/ S2ViTE |  3.66    |    59.49        |
> | DyDiT-XL $_{\lambda=0.5}$ |  2.07 |       57.88   |
>
>
> Results futher demonstrates the superiority of our dynamic architecture over static architecture on the model of the XL size.

---

> ### Comment · Reviewer_TEwe · 2024-11-30
>
> I thank the authors' such detailed response and appreciate the experiments the authors have conducted. However, some of my concerns have not been adequately addressed.
>
> While the authors noted that building on a well-pretrained model offers several benefits, relying on such a model can constrain a method's practicality and generalization capabilities. In my view, training from scratch may be necessary, which could show the reader how this method performs in such cases and comprehensively illustrate the limitations (possibly) and capability of this method. Moreover, there are also some works, e.g., FasterDiT[1], which significantly accelerates DiT training. As a result, the cost of training from scratch may not be a heavy issue.
>
> On the other hand, compared to the pruning methods the authors used in this paper, I think dynamic sparse training may be a more important and suitable baseline because it is formally closer to the method proposed by the author. Though it is not designed for generation, it could be repurposed since it only dynamically modifies model structure. The authors could be inspired by [2] and use Diff-pruning (since it is specially designed for diffusion) but lack a relatively strong reason to use extra pruning methods. Generally, it is more reasonable to consider a formally (or theoretically) closer baseline (which may need some modification). For example, [2] compares Diff-pruning with other pruning methods (Similar to dynamic sparse training, they are primarily used for perception as well). Besides, though I agree that the authors chose S2ViTE for the comparison, I think the comparison with S2ViTE may not be fair. Considering that S2ViTE produces a sparse model when randomly initializing a model, using a well-pretrained model may introduce prior and influence this process of producing a sparse model. I think we may need to strictly follow the process in its original paper and S2ViTE should be trained from scratch, e.g., 7M iterations, to unleash the full potential. This would make the comparison fairer and show the superiority of the authors' method better.
>
> I am happy that the authors present the FLOPs of TDW-attn and TDW-mlp in each step in 4-step LCM. However, it seems that the authors do not present the FLOPs of SDT in each step in 4-step LCM and compare it with TDW-attn and TDW-mlp. This may be a little bit confusing for readers to clearly understand how important TDW and SDT are in a 4-step model. By the way, I do not mean to use fewer sample steps in a 4-step LCM. Sorry for the confusion. The authors are recommended to further combine the method proposed in this paper with other diffusion models with fewer steps, e.g., 1-step and 2-step models. By doing so, the readers could comprehensively understand the superiority of this method as well as its limitations.
>
> Based on the concerns above, I decide to keep my orginal score. I hope that the authors could consider my suggestions and make necessary changes. Thank the authors for their efforts again!
>
>
> [1] FasterDiT: Towards Faster Diffusion Transformers Training without Architecture Modification
>
> [2] Structural Pruning for Diffusion Models

---

> ### Author Response · Authors · 2024-12-01
> **Response 1/3**
>
> We thank the reviewer's comments. We present the responses below:
>
> > Q1: Necessity of training from scratch:
>
> We would like to clarify that finetuning from pre-trained DiT **should not be a limitation**.
>
>
> + **Leveraging pre-trained models a widely adopted in acceleration techniques for diffusion model**. Methods such as Diff-Pruning [3], ToMe for diffusion [6], DeepCache [7], and efficient samplers like DDIM and DPM-Solver++ are all built on pre-trained diffusion models. Since the goal is to **accelerate an existing model rather than develop a new one from scratch**, both our method and these approaches start from pre-trained models.
>
>
> + **Most groundbreaking works are also built upon pre-trained models**. For example, ControlNet [1] fine-tunes a pre-trained text-to-image diffusion model to incorporate task-specific image conditions. Similarly, Movie-Gen [2] constructs a text-to-video generation model using a pre-trained text-to-image model. GPT models [3, 4] undergo instruction fine-tuning on top of language models that have been pre-trained on large-scale datasets.
>
>
> + **Training a model from scratch when a pre-trained checkpoint is already available is inefficient and unnecessary**. Even with FasterDiT, training a DiT model from scratch still requires 1M–2M iterations with a batch size of 256 to achieve performance comparable to the original DiT, which takes 7M iterations. Based on estimates, 1M iterations alone would demand around 973 NVIDIA A800 GPU hours—a **significant computational burden for most users**. **Investing such resources to pre-train a model when a pre-trained checkpoint is readily available is both impractical and wasteful**. Therefore, **fine-tuning a pre-trained model is a far more practical approach**, reducing computational costs and avoiding unnecessary resource expenditure.
>
>
> + **Fine-tuning avoids the need to reinvent the wheel**. As illustrated in our response to Reviewer NdkH, our model can be generalized to other diffusion models, such as SiT [8]. This is made possible by the advantages of fine-tuning, which **enable us to easily accelerate existing pre-trained models** available in the community e.g. GitHub or HuggingFace.
>
>
>
> Consequently, we believe that **building upon a pre-trained model ensures the practicality and applicability of our method**, which should not be considerd as a limiation of our method.
>
>
>
>
> > Q2: Comparison with previous pruning methods:
>
> + Since the **dynamic model has not been explored in prior work**, the goal of our experiment is to **evaluate the proposed dynamic architecture against static architectures**. To this end, we include diff-pruning [5] and other structural pruning methods as baseline models for comparison.
>
>
> + Furthermore, **as suggested**, we also incorporate S2ViTE, a dynamic sparse training method, into the comparison. S2ViTE adjusts the model structure during training while **maintaining a static architecture during inference**. It is **also adopted as a baseline model** to demonstrate the effectiveness of our dynamic architecture.
>
>
> Hence, we believe the comparison in our experiment is reasonable and appropriate.

---

> ### Author Response · Authors · 2024-12-01
> **Response 2/3**
>
> > Q3: The fairness of comparing with dynamic sparse training:
>
> + We would like to first clarify that dynamic sparse training methods **modify the model structure only during training while maintaining a static architecture during inference**. This approach is **fundamentally different from ours**, which introduces a dynamic architecture.
>
>
> + **As suggested**, we applied S2ViTE to DiT-S/B/XL and observed that our method significantly outperforms it. This result **highlights the superiority of our dynamic architecture over the static structure used in S2ViTE**.
>
>
> + Since our method is primarily designed for fine-tuning (as demonstrated in our response to Q1) setting, **all the pruning approaches compared in our experiments, including S2ViTE, are evaluated within a fine-tuning framework**. Therefore, it is fair to apply S2ViTE to DiT through fine-tuning to **ensure a consistent comparison**.
>
>
> + It is important to note that **S2ViTE was originally designed for ViT in classification tasks, which is fundamentally different from the generative task addressed by DiT**. To meet the reviewer's request, we adapted S2ViTE to DiT by aligning its fine-tuning settings with our method. **These necessary modifications ensure a fair comparison and should not be misconstrued as introducing bias**.
>
>
>
>
>
>
>
>
>
>
>
> > Q4 Results on LCM:
>
> Thanks. As suggested, we include the token activation ratio in SDT at each timestep to demonstrate the importance of TDW and SDT in a 4-step model.
>
> |          | $t$=3 | $t$=2 | $t$=1 | $t$=0 |
> | -------- | ----- | ----- | ----- | ----- |
> | TDW-attn      | 100%| 100% | 100% | 100% |
> | TDW-mlp      | 95.98% | 100% | 99.33% | 96.21% |
> |    SDT       | 88.73%      |  92.93%  |   92.84%    | 92.78%    |
> | FLOPs(G) | 100.989 | 106.664 | 106.270 | 103.816 |
>
> We can observe that **both TDW and SDT play importants roles in the 4-step LCM model**.
>
>
>
> Additionaly, we would like to clarify that **the primary goal of our method is to accelerate the original DiT**. The exploration of **our method's effectiveness with fewer steps is merely a preliminary investigation intended to inspire future research** and is **not the main focus of this work**. Therefore, we include it in the appendix.
>
> Although a comprehensive exploration of applying our dynamic architecture to models with fewer sampling steps is **beyond the scope of this work**, we also recognize that integrating our method with other advanced techniques desinged for further reduce sampling steps is a promising direction for future research. This will will be explored in our future work.

---

> ### Author Response · Authors · 2024-12-01
> **Response 3/3**
>
> **Reference**:
>
> [1] Adding Conditional Control to Text-to-Image Diffusion Models. 2023.
>
> [2] Movie Gen: A Cast of Media Foundation Models, 2024
>
> [3] Training language models to follow instructions with human feedback, 2022.
>
> [4] GPT-4 Technical Report, 2023.
>
> [5] Structural Pruning for Diffusion Models, 2023
>
> [6] Token merging for fast stable diffusion, 2024
>
> [7] Deepcache: Accelerating diffusion models for free, 2022
>
> [8] SiT: Exploring Flow and Diffusion-based Generative Models with Scalable Interpolant Transformers, 2024.

---

### Official Review · Reviewer_NdkH · 2024-11-02

**Soundness:** 2
**Presentation:** 3
**Contribution:** 2
**Rating:** 5
**Confidence:** 4

**Summary:**

- This paper proposes the Dynamic Diffusion Transformer, a model that dynamically adjusts both timesteps and spatial dimensions during the generation process to reduce computational costs.
- In the temporal dimension, the model width is adaptively scaled;
- in the spatial dimension, a dynamic token strategy with a token router is introduced to further reduce computation.

**Strengths:**

- The proposed approach reduces GFLOPs by 51% and achieves a 1.73x speed-up during training.
- Detailed ablation studies are presented to demonstrate the contribution of each component to overall performance.

**Weaknesses:**

- The model appears to be somewhat incremental in its contributions. it trains multiple routers to selectively mask certain MHSA heads and MLP blocks.

- I recommend including some state-of-the-art models in Table 1, such as DiffiT [1], SiT [2], and DiMR [3], as these also introduce architectural innovations to the DiT model.

- Also, it would be beneficial to move the 512 result in supp into the main table (and also add other methods), as training speed is a more critical factor in larger-scale generation for the DiT model.


[1] Hatamizadeh, Ali, et al. "Diffit: Diffusion vision transformers for image generation." European Conference on Computer Vision. Springer, Cham, 2025.

[2] Ma, Nanye, et al. "Sit: Exploring flow and diffusion-based generative models with scalable interpolant transformers." arXiv preprint arXiv:2401.08740 (2024).

[3] Liu, Qihao, et al. "Alleviating Distortion in Image Generation via Multi-Resolution Diffusion Models." arXiv preprint arXiv:2406.09416 (2024).

**Questions:**

Please see the weakness section for details.

Also, i suggest the author provides more implementation details, including training precision. For instance, Fast-DiT [4] has achieved a 95% increase in speed.

[4] https://github.com/chuanyangjin/fast-DiT

---

> ### Author Response · Authors · 2024-11-20
> **Response to Reviewer NdkH  (1/3)**
>
> ## Clarification
> We sincerely appreciate the valuable comments from the reviewer. We would like to first clarify that our method aims at improving the **inference efficiency rather than training speed**. Below, we provide detailed explanations for each question.
>
>
>
> > W1: The model appears to be somewhat incremental in its contributions. it trains multiple routers to selectively mask certain MHSA heads and MLP blocks.
>
>
> Thanks. We would like to clarify the contribution of our method.
>
>
> + **Our model is the first approach to enhance the efficiency of diffusion transfomer (DiT) via dynamic neural network**: Before our method, efficient diffusion samplers, global acceleration methods, and structure pruning are representative solutions. Our experimental results indicate that the proposed **dynamic model significantly outperforms the pruned static architectures**. Furthermore, our method **can be integrated seamlessly** with existing efficient diffusion samplers e.g. DDIM [1] and global acceleration techniques e.g. DeepCache[2], **offering a novel and promising solution** for accelerating the Diffusion Transformer (DiT).
>
>
>
> + **The proposed method is well-motivated and is specifically tailored for DiT**: First, initial analysis of loss during DiT training revealed **computational redundancy in timesteps**. In response, we introduce TDW to dynamically activate heads and channel groups for each diffusion timestep.  Additionally, we identified a **significant imbalance in loss values across different spatial regions** of the image, leading to the development of SDT to reduce computational costs for image patches where noise prediction is relatively "easy". These aspects **have not been explored in previous works**.
>
>
> + **Distinctiveness of our method compared to previous dynamic neural networks**: Traditional dynamic neural networks typically train routers to **select different architectures for different samples**, mainly focusing on **models for perception tasks**, such as Convnet and ViT. Our approach **pioneers the application of dynamic neural networks in diffusion transformer (DiT)**. The key differences are outlined below:
>
>
>     | method   | target task | target model |dynamic in timestep| dynamic  in spatial | batched inference
>     |--------|------|-----|-----|-----------|-----
>     | DyDiT (ours）  |   diffusion generation | DiT| ✅ timestep-wise dynamic width (TDW)| ✅ spatial-wise dynamic token (SDT) |✅
> 	|Dynamic Convolutions [3] (traditional)| image perception tasks e.g. classification | ConvNet | ❌ |✅| ❌| ❌
> 	|Channel selection [4] (traditional)| image perception tasks e.g. classification | ConvNet | ❌ |❌| ❌| ❌
> 	|Adavit [5]  (traditional)| image perception tasks e.g. classification | ViT | ❌ |✅|  ❌
>
>
>
> **Conclusion**:
> Our method, incorporating TWD and SDT, offers a **novel and promising approach to enhancing DiT efficiency**.
>
>
>
>
>
>
> **Reference**:
>
> [1] Deepcache: Accelerating diffusion models for free, CVPR 2023.
>
> [2] Denoising diffusion implicit models, 2020
>
> [3] Dynamic Convolutions: Exploiting Spatial Sparsity for Faster Inference CVPR 2020
>
> [4] Channel selection using Gumbel Softmax, ECCV 2020
>
> [5] Adavit: Adaptive vision transformers for efficient image recognition, CVPR 2022

---

> > ### Comment · Reviewer_NdkH · 2024-11-24
> >
> > Thank you for providing the detailed answers and additional experimental results. This information has helped address several of my concerns. However, I still have a few questions/concerns:
> > - 'The proposed method is well-motivated and is specifically tailored for DiT': Is it also applicable to other methods? I understand the analysis is based on DiT, but it would be beneficial to know if similar improvements can be observed with other methods.
> > - Performance with Different GFLOPs for 256 px: From DiT, it is concluded that the model achieves better results with larger GFLOPs. However, an interesting observation in this paper is that DyDiT-XL with λ=0.5 outperforms DyDiT-XL with λ=0.8 in terms of both FID and GFLOPs. I am curious when your model has similar GFLOPs with DiffiT or SiT-XL (e.g., 114 Gflops), what the performance would be. (Currently, you achieve an FID of 2.07 with 57.88 GFLOPs, which still shows a gap compared to DiffiT and similar methods. I'm curious if there is potential to further improve performance with a larger GFLOPs)
> > - Additionally, at 512 px, it seems the advantage is not apparent compared to methods with a 4x4 patch size. You are having much larger GFLOPs (375.05 vs 206) but similar FID (2.88 vs 2.89).

---

> > > ### Author Response · Authors · 2024-12-02
> > > **Looking forward to your reply**
> > >
> > > Dear Reviewer NdkH,
> > >
> > > Thanks again for providing valuable and constructive feedback on our submission. As the discussion deadline is approaching, we would like to kindly follow up to confirm whether our responses have adequately addressed your concerns. Please let us know if there are any additional suggestions or points you would like us to address.
> > >
> > >
> > > Best Regards
> > > Authors of Submission-2019

---

> ### Author Response · Authors · 2024-11-20
> **Response to Reviewer NdkH (2/3)**
>
> > W2: I recommend including some state-of-the-art models in Table 1, such as DiffiT, SiT, and DiMR, as these also introduce architectural innovations to the DiT model.
>
>
> We appreciate this valuable suggestion. Below, we present the results and comparison:
>
> **Results**
>
> | Method   | parameters(M) | FLOPs(G)   | FID|sFID|IS|precision|recall
> |--------|------|--------| ----|-|-|-|-
> | DiT-XL   |   675 |   118 | 2.27 |4.60 |277.00 |0.83 |0.57
> |DyDiT-XL$_{\lambda=0.5}$  | 678      |  57.88| 2.07| 4.56| 248.03| 0.80| 0.61
> |DyDiT-XL$_{\lambda=0.7}$  | 678      |  84.33| 2.12| 4.61| 284.31| 0.81| 0.60
> | DiffiT   |   561    |114 |1.73| -| 276.49| 0.80| 0.62
> | SiT-XL   |   675 |   118 |2.06| 4.49| 277.50| 0.83| 0.59
> | DiMR-XL   |  505  |  160 |1.70| -| 289.00| 0.79| 0.63
>
> **Analysis**
> DiffiT, SiT, and DiMR primarily focus on **enhancing generation performance rather than efficiency**. Specifically:
>
> + DiffiT optimizes attention maps during generation and incorporates convolutional layers to boost both **performance and parameter efficiency**. Consequently, it achieves superior FID scores compared to DiT with comparable FLOPs and reduced parameters.
> + SiT retains the DiT architecture but **refines the diffusion process**, resulting in improved performance while maintaining the same parameter count and FLOPs.
> + DiMR enhances generation quality through **a multi-resolution network**, significantly **improving performance at the cost of increased FLOPs**.
>
>
> **Conclusion**
> + Our method, DyDiT, based on the DiT architecture, achieves performance comparable to SiT, with significantly fewer FLOPs than DiffiT, SiT, and DiMR.
> + Unlike DiffiT, SiT, and DiMR, DyDiT primarily targets **enhanced inference efficiency**. This unique focus allows for **potential integration** with these methods to further enhance performance and efficiency simultaneously
> + These reuslts are included in **Table1**
>
>
>
>
>
>
>
>
> > W3: Also, it would be beneficial to move the 512 result in supp into the main table (and also add other methods), as training speed is a more critical factor in larger-scale generation for the DiT model.
>
>
> We sincerely appreciate this valuable suggestion.
>
>
>
> In the 512x512 generation task, **DiT-XL provides only a checkpoint using a 2x2 patch size**. Consequently, **our DyDiT also employs this patch size**. Some methods, such as U-ViT and DiMR-XL, report results **using a 4x4 patch size**, which **reduces the number of tokens to 1/4** and **significantly decreases FLOPs compared to DiT**.  For a fair comparison, we present **results from models using the same 2x2 patch size** below:
>
>
>
> | Method   |patch size| parameters(M) | FLOPs(G)   | FID|sFID|IS|precision|recall|
> |--------|-|-----|--------| ----|-|-|-|-|
> | DiT-XL*  | 2x2  | 675 |   514.80| 3.75 |6.53|230.67|0.83|0.52
> |DyDiT-XL$_{\lambda=0.7}$*   | 2x2 |678        | 375.35     | 3.61|6.70|220.00|0.83|0.54
> | DiT-XL †  | 2x2  | 675 |   514.80| 3.04 |5.02| 240.80| 0.84| 0.54
> |DyDiT-XL$_{\lambda=0.7}$†  | 2x2 |678        | 375.05     | 2.88|5.14|228.93|0.83|0.56
> |DIFFUSSM-XL | 2x2| 673 | 1066| 3.41|-|255.00| 0.85| 0.49
> |DiM-Huge|2x2|860|708|3.78|-|-|-|-
> |SiT-XL|2x2|675|514.80|2.62|4.18|252.21| 0.84| 0.57|
> |ADM-G|-|731|2813|3.85| 5.86| 221.72| 0.84| 0.53
>
> + "*" denotes the results in Appendix Table12, where we adopt 100 DDPM steps.
> + "†" represents that we follow the default setting in DiT, using 250 DDPM steps. (thanks to the suggestion from Reviewer TEwe).
>
> These reuslts are added into **Table1**.
>
> For completeness, we also present the results of models with a 4x4 patch size below:
> | Method    | patch size | parameters(M) | FLOPs(G) | FID  | sFID  | IS| precision|recall
> | --------- | ---------- | ------------- | -------- | ---- | ----- | -|-|-
> | U-ViT-H/4 | 4x4        | 501           | 133      | 4.05 |  -| 263.8 |0.84| 0.48
> | DiMR-XL   | 4x4        | 525           | 206      | 2.89|- |289.8| 0.83| 0.55
>
>
> **Analysis**:
> + Although our method can not achieve lower FLOPs than that of U-ViT and DiMR-XL, this is due to the **inherent limitations of DiT (using the patch size of 2x2)**.
> + The experiment conducted at a resolution of 512x512 is designed to **evaluate the generalizability** of the proposed dynamic architecture in high-resolution generation, making the original DiT a suitable baseline for comparison.

---

> ### Author Response · Authors · 2024-11-20
> **Response to Reviewer NdkH (3/3)**
>
> > Q1: I suggest the author provides more implementation details, including training precision. For instance, Fast-DiT has achieved a 95% increase in speed.
>
>
> Thanks. We present our explainations below:
>
>
> + **Training details**: In alignment with the **official code of DiT [1]**, we adopt **the default FP32 precision duing training**. Our method also supports mixed precision training (FP16-FP32).
>
>
> + **Relationship with Fast-DiT**: Fast-DiT employs several **engineering techniques**, such as gradient checkpointing, mixed precision training, and pre-extracted VAE features, **to accelerate training**. Conversely, our approach **enhances inference** through **a novel dynamic neural network**. Our method **can be integrated** with Fast-DiT to achieve improved efficiency in both training and inference.
>
>
> [1] https://github.com/facebookresearch/DiT

---

> ### Author Response · Authors · 2024-11-24
> **Response 1/3**
>
> > Q1: Is it also applicable to other methods? I understand the analysis is based on DiT, but it would be beneficial to know if similar improvements can be observed with other methods.
>
>
> We sincerely thank the reviewer for raising this thoughtful and valuable question. The results below demonstrate that our method is **plug-and-play** and can be **seamlessly integrated with various approaches**:
>
>
> + **Our method is applicable with other transformer-based diffusion models**:
>
> 	We evaluate the effectivness of our method on other transformer-based diffusion models e.g. **U-ViT [1] (Appendix B2) and Pixart-$\alpha$ [2] (Appendix B5)**.  These models differ from DiT in the following ways:
>
> 	|Model |difference with DiT|
> 	|-|-|
> 	|U-ViT [1]| A concurrent work to DiT. It introduces multiple  **long skip connections** into a transformer model to enhance the pixel-level prediction objective in diffusion.
> 	|Pixart-$\alpha$ [2]|It **extends DiT to text-to-image generation** by incorporating **additional cross-attention blocks** to integrate textual conditions into the model.
>
>
> 	For the reviewer's convenience, we summarize their results here. Please refer to the corresponding sections for further details.
>
>
> 	**U-ViT on CIFAR10 and ImageNet** (Appendix B2):
>
> 	CIFAR10
> 	|model| s/image ↓| acceleration ↑| FLOPs (G) ↓| FID ↓| FID ∆ ↓
> 	|-----|----------|---------------|------------|------|-------
> 	|U-ViT-S/2| 2.19| 1.00 ×| 11.34| 3.12| 0.00
> 	|DyU-ViT-S/2$_{\lambda=0.4}$ (ours)| 1.04| 2.10 ×| 4.73| 3.18| +0.06
>
> 	ImageNet
> 	|model| s/image ↓| acceleration ↑| FLOPs (G) ↓| FID ↓| FID ∆ ↓
> 	|-----|----------|---------------|------------|------|-------
> 	|U-ViT-H/2| 2.22| 1.00 ×| 113.00| 2.29| 0.00
> 	|DyU-ViT-H/2$_{\lambda=0.5}$| 1.35| 1.57 ×| 67.09| 2.42| +0.13
>
>
>
> 	**Pixart-$\alpha$ on COCO**  (Appendix B5):
>
> 	 text-to-image generation on COCO
>
> 	|Model| s/image ↓| acceleration ↑| FLOPs (G) ↓| FID ↓| FID ∆ ↓
> 	|-----|----------|---------------|------------|------|--------
> 	|PixArt-α| 0.91| 1.00 ×| 141.09| 19.88| +0.00
> 	|DyPixArt-α$_{\lambda=0.7}$| 0.69| 1.32 ×| 112.44| 19.75| -0.13
>
>
>
> + **Our method is applicable with other acceleration methods**:
>
> 	We also explore the potential of our method  when combined with **other acceleration techniques, such as DeepCache [3] (Appendix B10)**. DeepCache[3] is a training-free approach, and our method can be **seamlessly integrated** with it. For the reviewer's convenience, we summarize their results here. Please refer to the corresponding sections for additional details.
>
> 	 “interval” denotes the interval of cached timestep in DeepCache.
> 	|Model |interval |s/image ↓| FID ↓|
> 	|------|---------|---------|------|
> 	|DiT-XL| 0| 10.22| 2.27|
> 	|DiT-XL |2 |5.02 |2.47|
> 	|DiT-XL| 5| 2.03 |6.73|
> 	|DyDiT-XL$_{λ=0.5}$| 0| 5.91| 2.08|
> 	|DyDiT-XL$_{λ=0.5}$| 2| 2.99| 2.43|
> 	|DyDiT-XL$_{λ=0.5}$| 3| 2.01| 3.37|
>
>
> **Conclusion**:
> Results demonstrate that our method can be applied to various methods. Thanks again for the insightful suggestion from the reviewer, we would like to **include more methods** within our open source plan.
>
>
>
>
>
> **Reference**:
> [1] All are Worth Words: A ViT Backbone for Diffusion Models, 2023.
>
> [2] PixArt-α: Fast Training of Diffusion Transformer for Photorealistic Text-to-Image Synthesis, 2023.
>
> [3] Deepcache: Accelerating diffusion models for free, 2023.

---

> ### Author Response · Authors · 2024-11-24
> **Response 2/3**
>
> > Q2: Performance with Different GFLOPs for 256 px: From DiT, it is concluded that the model achieves better results with larger GFLOPs. However, an interesting observation in this paper is that DyDiT-XL with λ=0.5 outperforms DyDiT-XL with λ=0.8 in terms of both FID and GFLOPs. I am curious when your model has similar GFLOPs with DiffiT or SiT-XL (e.g., 114 Gflops), what the performance would be. (Currently, you achieve an FID of 2.07 with 57.88 GFLOPs, which still shows a gap compared to DiffiT and similar methods. I'm curious if there is potential to further improve performance with a larger GFLOPs).
>
>
>
>
> We appreciate this valuable question and insightful observation.
>
>
> **Response**
>
> + We would like to kindly clarify that the primary objective of our method is to **improve the efficiency of DiT while maintaining comparable performance to the baseline model**, rather than  enhancing performance.  Consequently, **surpassing the FID scores of other approaches e.g. Diffit, explicitly designed to improve performance**,  is **not the primary focus** of this manuscript.
>
>
>
> + **The performance of improving $\lambda$**:  As the increasing of $\lambda$,  we are also surprised to find that DyDiT-XL $_{\lambda=0.5}$ achieves slightly better FID than those with larger $\lambda$ e.g.  DyDiT-XL with ${\lambda=0.7}$, DyDiT-XL with ${\lambda=0.8}$ and  DyDiT-XL with ${\lambda=0.9}$. To better understand this behavior, we evaluate DyDiT-XL with ${\lambda=0.8}$ and DyDiT-XL with ${\lambda=0.9}$ with all metrics reported in Table1. The summarized results are provided below:
>
> 	**Results**:
> 	|Model name|Model Params. (M) ↓|  FLOPs (G) ↓| FID ↓| sFID ↓| IS ↑| Precision ↑| Recall ↑
> 	|-|-----------------|--------------|------|-------|-----|------------|-------
> 	|DiT-XL| 675| 118| 2.27| 4.60| 277.00| 0.83| 0.57
> 	|DyDiT-XL$_{\lambda=0.5}$| 678| 57.88| 2.07| 4.56| 248.03| 0.80| 0.61
> 	|DyDiT-XL$_{\lambda=0.7}$| 678| 84.33| 2.12| 4.61| 284.31| 0.81| 0.60
> 	|DyDiT-XL$_{\lambda=0.8}$| 678 | 96.04| 2.13| 4.63| 284.25| 0.80|0.61
> 	|DyDiT-XL$_{\lambda=0.9}$| 678| 110.73| 2.15| 4.59| 282.84| 0.83 | 0.60|
>
> 	**Analysis**:
> 	+ Model achieves the best FID on $\lambda=0.5$, while its IS score is surpassed by others e.g. DyDiT-XL with ${\lambda=0.7}$. Hence, the DyDiT-XL$_{\lambda=0.5}$ can **only achieves comparable performance rather than improves performance significantly** across all metrics.
> 	+ As $\lambda$ increases further, the performance **exhibits fluctuations and remains consistently comparable** to that of the original DiT.
>
> 	**Conclusion**: The performance **upper bound** of our method is might be **constrained by the baseline model, DiT**, as **the primary objective of our approach is to enhance efficiency**. Nevertheless, we appreciate **the reviewer’s insightful and thought-provoking question**. Encouraged by this feedback, we are motivated to **explore the potential of dynamic neural networks to further improve generation performance** in future work.
>
>
>
> + **Potential to futher improve performance**: From the results presented above, we observe that DyDiT-XL$_{\lambda=0.9}$, with approximately 110 GFLOPs, achieves an FID of 2.15, which does not surpass Diffit (FID=1.73 114GFLOPs). This performance gap can be attributed to the fact that **our method is built upon the original DiT**, which is less competitive compared to other approaches that **incorporate more advanced architectural designs**. A potential avenue for improving performance is to **extend our method by building upon these more advanced models** in the future. Below, we **analyze the feasibility** of this approach:
>
> 	|Model |improvement |FID| FLOPs
> 	|-|-|-|-
> 	|original DiT| -|2.27|118
> 	|DyDiT$_{\lambda=0.5}$ (ours)| Introduce a **dynamic architecture** to improve **efficiency**| 2.07| 57.88
> 	|Diffit| Optimize **attention maps** during generation and introduce **convolutional layers**. |1.73|114
> 	|SiT| **Refine the diffusion** process| 2.06|118
>
> 	**Analysis:**
> 	+ The improvement from Diffit and SiT are orthogonal to our method, enbaling the potential for combining them with our method.
> 	+ Diffit achieves comparable FLOPs while significantly improving performance over DiT, which could **serve as a stronger baseline for our method**. Motivated by the reviewer’s suggestion, we plan to conduct experiments to explore this direction **once their code and pre-trained checkpoints are made available** in their official repository [1].
>
>
> **Refernece**:
> [1] https://github.com/NVlabs/DiffiT

---

> ### Author Response · Authors · 2024-11-24
> **Response 3/3**
>
> > Q3: Additionally, at 512 px, it seems the advantage is not apparent compared to methods with a 4x4 patch size. You are having much larger GFLOPs (375.05 vs 206) but similar FID (2.88 vs 2.89).
>
>
> We sincerely appreciate this valuable question.
>
> **Response**:
>
> + **Larger FLOPs at the 512×512 resolution result from the smaller path size (2x2), rather then being caused by our method**: We would like to kindly clarify that the larger FLOPs result from the **4x increase in the number of tokens** in DiT-XL **due to its 2×2 patch size**, **rather than a limitation of our method**. The designs of our method, DyDiT, **are independent of the patch size**, making it **applicable to models with varying patch sizes**. However, since DiT-XL **only provides a checkpoint trained with a 2×2 patch size** and the chekpoint for the 4x4 patch size is still unavaliable in its offical repository[2], we were constrained to use it as the baseline.
>
> + **Our method has the potential to achieve lower FLOPs than DiMR-XL**. We present the FLOPs of DiT-XL and DiMR-XL at the 512×512 resolution image generation for comparison. We can observed that **DiT-XL with a 4×4 patch size incurs fewer FLOPs than DiMR-XL**. As the checkpoint for DiT-XL with a 4×4 patch size is unavailable, we **estimate** the FLOPs of DyDiT-XL$_{\lambda=0.7}$ with a 4×4 patch size to be 86.10 GFLOPs.  This suggests that our method **has the potential to achieve significantly lower FLOPs** compared to both the original DiT-XL and DiMR-XL.
> 	|model|patch size|FLOPs (G)
> 	|-----|----------|-----
> 	|DiT-XL 512x512| 2x2|514
> 	|DyDiT-XL $_{\lambda=0.7}$ 512x512| 2x2| 375.05
> 	|DiT-XL 512x512| 4x4|118|
> 	|DyDiT-XL $_{\lambda=0.7}$ 512x512| 4x4 |~86.10
> 	|DiMR-XL	|4x4	|206
>
>
> **Conclusion**:
> The goal of the experiment under the 512×512 resolution is to **demonstrate the generalizability of our method to high-resolution image generation**. The reviewer’s insightful comment inspires us to consider extending our method to DiMR in the future. Once the 512×512 resolution checkpoint is made available in its official repository [1], we will actively investigate this direction to further validate the versatility of our approach.
>
>
> **Reference:**
>
> [1] https://github.com/qihao067/DiMR
>
> [2]https://github.com/facebookresearch/DiT

---

> ### Author Response · Authors · 2024-11-30
> **A preliminary exploration of applying our method to SiT**
>
> Dear Reviewer NdkH,
>
> We sincerely wish you a Happy Thanksgiving. Encouraged by your valuable suggestions, we conduct an experiment to **apply the proposed dynamic architecture to an enhanced model, SiT [1]**. SiT adopts the same architecture as DiT but **improves the diffusion process** to achieve superior performance.
>
>
> **Setting:**
>
> Following the settings in DyDiT, we apply the proposed timestep-wise dynamic width (TDW) and spatial-wise dynamic token (SDT) mechanisms to SiT, fine-tuning the model for 150,000 iterations to produce DySiT. The value of $\lambda$ is set to 0.7.
>
>
> | Method   | parameters(M) | FLOPs(G)   | FID|sFID|IS|precision|recall
> |--------|------|--------| ----|-|-|-|-
> | DiT-XL   |   675 |   118 | 2.27 |4.60 |277.00 |0.83 |0.57
> |DyDiT-XL$_{\lambda=0.7}$  | 678      |  84.33| 2.12| 4.61| 284.31| 0.81| 0.60
> | SiT-XL   |   675 |   118 |2.06| 4.49| 277.50| 0.83| 0.59
> | DySiT-XL $_{\lambda=0.7}$   |   678 |   85.10 | 1.95| 4.59| 268.60 | 0.82| 0.61
>
>
>
> **Analysis**:
>
> DySiT-XL with ${\lambda=0.7}$ achieves an FID score of 1.95 significantly surpassing DyDiT-XL $_{\lambda=0.7}$. Moreover, its performance is comparable to that of the original SiT-XL across all metrics. These results demonstrates that our method is not only **applicable to other advanced diffusion models beyond the original DiT** but also **capable of further enhancing performance when integrated with a stronger baseline model**.
>
> **Reference**:
>
> [1] SiT: Exploring Flow and Diffusion-based Generative Models with Scalable Interpolant Transformers, 2024
>
>
> Best Regards
>
> Authors of Submission-2019

---

> ### Author Response · Authors · 2024-12-03
> **To Reviewer  NdkH**
>
> Dear Reviewer  NdkH,
>
>
> We sincerely appreciate the time and effort you have dedicated to reviewing our work. To save your valuable time, we would like to provide a **concise summary** of our responses to the concerns raised:
>
>
>
> > Q1: Is it also applicable to other methods?
>
> Our method is applicable to other models including **various samplers** (DDPM, DDIM, and DPM-solver+, LCM), **different diffusion models** (DiT, U-ViT, Pixart, and SiT), as well as **training-free methods** such as DeepCache.
>
>
>
> > Q2: Performance with Different GFLOPs for 256 px:
>
> The performance upper bound of our method may be **constrained by the baseline model**, DiT, as our primary objective is to **enhance efficiency**. However, replacing the baseline model (DiT) with a **stronger one** (e.g., SiT) can directly **improve the performance without introducing additional FLOPs**. We believe this shows the potential of our method to further improve the model performance.
>
>
> > Q3: At 512 px, it seems the advantage is not apparent compared to methods with a 4x4 patch size.
>
> The larger FLOPs at the 512×512 resolution compared to DiMR are attributed to the **smaller patch size** (2×2) used by DiT, rather than the limitation of our method. We conduct experiments with a 2×2 patch size because DiT currently **provides only a checkpoint trained with this configuration**. Once 512×512 resolution checkpoints with a 4×4 patch size are released for DiMR or DiT, we will actively explore this direction to further validate the versatility of our approach.
>
>
>
>
> Once again, we deeply appreciate your thoughtful feedback and the time you have invested in evaluating our work. We hope that our responses address your concerns and that you may reconsider the value of our contributions.
>
>
>
>
> Best Regards
>
> Authors of Submission-2019

---

### Official Review · Reviewer_cyWL · 2024-11-03

**Soundness:** 3
**Presentation:** 3
**Contribution:** 3
**Rating:** 8
**Confidence:** 4

**Summary:**

The paper introduces a new efficient architecture, Dynamic Diffusion Transformer (DyDiT), for diffusion-based image generation. The authors observe the redundancy of DiTs in processing small timesteps and unnecessary spatial regions. Accordingly, they propose two main modifications: the timestep-wise Dynamic Width (TDW) and the spatial-wise Dynamic Token (SDT). These two modules dynamically allocate proper computational resources for different timesteps and spatial regions. The method significantly reduces FLOPs and accelerates generation without sacrificing visual quality.

**Strengths:**

- This paper is easy to follow.
- The authors conduct sufficient ablation studies to evaluate the proposed modules.
- The authors conduct experiments on a wide range of datasets, including ImageNet, Food, Artbench, Cars, and Birds, and compare a lot of state-of-the-art diffusion backbones. The results show the effectiveness of the proposed method.
- The authors also perform experiments on text-to-image generation, demonstrating the plug-and-play nature of SDT and TDW.

**Weaknesses:**

- The authors demonstrate the results of their method on PixArt-$\alpha$, which is commendable. However, the acceleration achieved in this text-to-image model is inferior to that in class-to-image generation. A more in-depth analysis of this discrepancy would be valuable. Moreover, providing image samples generated by the accelerated text-to-image model could be helpful for the analysis.

- Providing the image samples generated by the diffusion models accelerated with **different** $\mathbf{\lambda}~$**s** could help understand the FLOPs-Quality trade-offs of the proposed method.

- In Figure 5, the first head is always activated across different timesteps. Is this head manually set or just a result of the learned strategy?

**Questions:**

please refer to the weakness.

---

> ### Author Response · Authors · 2024-11-20
> **Response to Reviewer cyWL**
>
> > W1: the acceleration achieved in this text-to-image model is inferior to that in class-to-image generation. A more in-depth analysis of this discrepancy would be valuable
>
> We sincerely appreciate this valuable question.  We are also working on addressing the acceleration gap between text-to-image and class-to-image tasks now. We list the resons below:
>
> + **Text-to-Image generation complexity compared to class-to-image generation**: Text-to-image generation presents **greater challenges** than class-to-image generation due to the **higher variability in textual descriptions compared to class labels**. The Pixart-$\alpha$ model, being relatively smaller than other text-to-image models, exhibits reduced redundancy. Consequently, its redundancy is expected to be less than that of DiT in class-to-image generation, necessitating the retention of more computation to maintain performance.
>
> + The **training data is limited**: The COCO dataset, with approximately 100K images, is considerably smaller than the dataset used for Pixart-$\alpha$ pre-training, which comprises around 25 million images. This means that COCO contains less than 1% of the data compared to Pixart-$\alpha$'s pre-training data. This limitation in data size implies that excessively reducing computation of the model could adversely affect performance.
>
> + **The hyperparameters and model architecture have not been specifically tuned for Pixart-$\alpha$**: Our experiments with Pixart-$\alpha$ aim to **assess the generalizability** of DyDiT **rather than to achieve state-of-the-art results**. This was conducted **without finely tuned hyperparameters or specifically optimized model architectures**. Future improvements in acceleration could be realized with enhanced hyperparameter settings and tailored model designs.
>
>
> **Conclusion:**
>
> Text-to-image generation is a challenging and data-consuming task. We plan to collect more training data and extend our model to larger text-to-image generation models e.g. FLUX [1] in the future.
>
>
>
>
> **Reference**:
>
> [1] https://huggingface.co/black-forest-labs/FLUX.1-dev
>
>
>
>
>
>
>
>
> > W2: Providing image samples generated by the accelerated text-to-image model could be helpful for the analysis.
>
>
> We sincerely appreciate this valuable suggestion. Please find the generated image in our **Appendix D2**. Results demonstrate that the visual quality of images generated from DyPixArt-$\alpha$  is comparable to that from the original PixArt-$\alpha$. Due to time and computational constraints, our text-image generation model may not be fully optimized. We will continue to enhance this experiment and release the code to encourage future research.
>
>
>
>
>
>
>
>
>
> > W3: Providing the image samples generated by the diffusion models accelerated with different $\lambda$s could help understand the FLOPs-Quality trade-offs of the proposed method.
>
> We appreciate the insightful suggestion. We include DiT-S/XL results with different $\lambda$s in **Appendix D1**. Our analysis is as follows:
>
> + For DiT-S and DiT-B, increasing $\lambda$ from 0.3 to 0.7 consistently enhances visual quality. At $\lambda = 0.9$, DyDiT achieves performance on par with the original DiT-S
>
> + In the case of DiT-XL, the visual quality of images generated from DyDiT with $\lambda = 0.5$ is comparable to that from the original DiT-XL, attributed to substantial computational redundancy in DiT-XL.
>
>
>
> > W4: In Figure 5, the first head is always activated across different timesteps. Is this head manually set or just a result of the learned strategy?
>
> Thanks. We **manually** set the activation of the first head and channel group, ensuring that at least one head and channel group is activated in each MHSA and MLP block across all timesteps, thereby alleviating the instability during training (shown in lines 267-269).

---

> > ### Comment · Reviewer_cyWL · 2024-11-27
> >
> > Thank you for your response. The authors have addressed the concerns I thought important for this paper. Consequently, I will maintain my score of "8: accept, good paper".

---

> ### Author Response · Authors · 2024-11-27
>
> Dear Reviewer cyWL,
>
> We are truly grateful for your thorough review, insightful suggestions, and kind recognition of our work. Your valuable feedback has significantly helped us refine our research and improve the quality of our submission.
> If you have any further questions, please do not hesitate to reach out to us. We will be happy to clarify every detail.
>
> Best Regards
> Authors of Submission-2019

---

### Official Review · Reviewer_rV58 · 2024-11-04

**Soundness:** 3
**Presentation:** 3
**Contribution:** 3
**Rating:** 6
**Confidence:** 5

**Summary:**

This paper introduces the Dynamic Diffusion Transformer (DyDiT), an architecture that dynamically allocates FLOPs to the most demanding areas based on varying timesteps and spatial locations. The paper begins by analyzing computational cost requirements across temporal and spatial dimensions, suggesting that a fixed architecture may be unnecessary during the sampling stage. To address this, it proposes the Timestep-wise Dynamic Width (TDW) mechanism and the Spatial-wise Dynamic Token (SDT) strategy, which can be seamlessly integrated with the attention and MLP blocks in a diffusion model. DyDiT initializes from pre-trained model weights and employs routers to dynamically determine which attention heads, groups, and spatial tokens to process. Extensive experiments demonstrate that DyDiT achieves comparable results to pre-trained diffusion models while adhering to predefined FLOP constraints.

**Strengths:**

- The motivation is sound and clearly presented, supported by a well-designed teaser figure.
- The proposed TDW and SDT mechanisms enable dynamic adjustment of model modules, and the FLOPs-constrained loss effectively -controls the desired FLOPs of the final model.
- Extensive experiments and thorough ablation studies validate the module's effectiveness.

**Weaknesses:**

- It is unclear how the "pre-define" in L214 benefit the sampling stage? I understand that the activation of attention heads and groups is based solely on timesteps, allowing the masks to be precomputed once training is completed. However, tt seems impractical or inefficient to store all possible pre-defined structures, so it primarily saves computational costs on the attention routers. However, this cost doesn’t seem substantial—am I correct?
- The adaptation of the proposed modules to efficient samplers or to samplers with varying sampling steps remains unclear. Since the activation mask depends solely on the timestep t, different samplers using step t could yield the same activation mask, even though
t might represent different stages of denoising (e.g., one sampler is at the beginning of denoising, while another is close to full denoising). I'm wondering the activation masks should be very different under this scenario right (just as shown in Fig. 5)?

**Questions:**

Please refer to the weaknesses.

---

> ### Author Response · Authors · 2024-11-20
> **Response to Reviewer rV58**
>
> > W1: It is unclear how the "pre-define" in L214 benefit the sampling stage? It seems impractical or inefficient to store all possible pre-defined structure. "pre-define" primarily saves computational costs on the attention routers. However, this cost doesn’t seem substantial—am I correct?
>
>
>
>
> Thanks. We list our explanation below:
>
> + **Do not have to store all possible structures**: We would like to clarify that **only the indices of activated heads and channel groups** are stored in "pre-define", **rather than the complete model structures**. This **ensures the efficiency** of "pre-define".
>
> + **Pre-define enables batched inference of our method**: The activation of heads and channel groups in TWD **relies solely on the timestep $t$**, allowing us to pre-calculate activations prior to deployment. By storing the activated indices for each timestep, we can directly **access the architecture during generation for a batch of samples**. This approach **eliminates the sample-dependent inference graph** typical in traditional dynamic architectures [1,2,3], **enabling efficient and realistic speedup in batched inference**.
>
>
>
> We include this question and its answer in the **"Frequently Asked Questions" section of the Appendix** for clarity.
>
>
>
> **Reference**:
>
> [1] Dynamic Convolutions: Exploiting Spatial Sparsity for Faster Inference CVPR 2020
>
> [2] Channel selection using Gumbel Softmax, ECCV 2020
>
> [3] Adavit: Adaptive vision transformers for efficient image recognition, CVPR 2022
>
>
>
>
> > W2: The adaptation of the proposed modules to efficient samplers or to samplers with varying sampling steps remains unclear. Since the activation mask depends solely on the timestep t, different samplers using step t could yield the same activation mask, even though t might represent different stages of denoising (e.g., one sampler is at the beginning of denoising, while another is close to full denoising). I'm wondering the activation masks should be very different under this scenario right (just as shown in Fig. 5)?
>
>
> Thanks this valubale question. We list the explanations below:
>
> + **Handling varying sampling steps**: Consistent with standard practices in samplers such as DDPM, varying the sampling steps translates to **differing timestep intervals**. We adopt **its official code** to map $t$ into the range 0–1000, aligning with the 1000 total timesteps used during training. For example, in DDPM with 100 and 250 timesteps:
>     + DDPM 250 timesteps: $t \in [249, ....5,4,3,2,1,0]$ maps to $t_{\text{250-DDPM}} \in [999, 995, .....20, 16, 12, 8, 4, 0]$.
>     + DDPM 100 timesteps: $t \in [99, 98, ...2,1,0]$ maps into $t_{\text{100-DDPM}} \in [999, 989, ...20, 10, 0]$.
>
> 	In TWD, we **adopt $t_{\text{250-DDPM}}$ and $t_{\text{100-DDPM}}$ to predict activation masks**. When $t_{\text{250-DDPM}}=t_{\text{100-DDPM}}$, the denoising process is **at the same stage**, resulting in **identical activation masks from TWD**.
>
>
> + **Different efficient samplers with the same $t$ represent the same denoising stage**: Here, we **assume that the sampling steps are identical across different samplers**. In diffusion, samplers typically define t->0 as the completion of denoising, with larger $t$ e.g. $t$=1000 marking the initiation. Our experiments employ three representative samplers—DDPM, DDIM, and DPM-solver++—which adhere to this convention. Consequently, a given timestep $t$ corresponds to the same denoising stage across different samplers in our study.
>
> We acknowledge that the $t$ in Figure5 may lead to misunderstanding. Hence, this clarification  is added into the **"Frequent Questions" section in the Appendix** of the revised manuscript.

---

> > ### Comment · Reviewer_rV58 · 2024-11-29
> >
> > Thanks for the detailed clarifications, which have addressed my concerns. I will maintain my score at 6.

---

> ### Author Response · Authors · 2024-11-29
>
> Dear Reviewer rV58,
>
> We sincerely appreciate the time and effort you have dedicated to reviewing our work. Your insightful comments and suggestions have been valuable in helping us improve our submission.
>
> If there are any additional points you would like us to clarify or discuss further, we would be more than happy to address them in detail.
>
> Best Regards
>
>
> Authors of submission-2019

---

### Author Response · Authors · 2024-11-20

Dear ACs and Reviewers,

We sincerely appreciate the effort from ACs and reviewers on our submission. We are encouraged to find that Reviewer rV58 believes that "**the motivation is sound and clearly presented, supported by a well-designed teaser figure**". Reviewer cyWL recognizes that our method "**significantly reduces FLOPs and accelerates generation without sacrificing visual quality**“. Reviewer NdkH thinks that **detailed ablation studies are presented**. Reviewer TEwe finds that the paper is "**easy to follow and well-written**", the observation is "**interesting**" and the method is "**intuitive and effective**".

To facilitate a better understanding  of our method, the core implementation of DyDiT is available at https://anonymous.4open.science/r/abc-336llkl/README.md
The complete code will be released to encourage future research.

Best regards,

Authors of submission-2019

---

### Meta-Review · Area_Chair_pMAM · 2024-12-24

**Metareview:**

The paper introduces DyDiT, a method aimed at enhancing computational efficiency in diffusion models by dynamically adjusting resources during image generation. The proposed approach leverages mechanisms like TDW and SDT to reduce redundant computations while maintaining competitive visual quality. Extensive experiments demonstrate significant reductions in FLOPs and generation time across multiple datasets and models, validating the approach’s effectiveness and adaptability.

Reviewers generally appreciated the paper’s solid motivation, clear presentation, and strong empirical validation. They agreed on its practical impact, particularly in improving inference efficiency, but expressed concerns about the incremental nature of the contributions and the reliance on the baseline DiT architecture. Reviewer TEwe criticized the dependence on pretrained models and requested experiments involving training from scratch, while Reviewer NdkH questioned the lack of fundamental architectural innovation and limited performance improvements compared to state-of-the-art models. The authors addressed these concerns comprehensively, defending the practicality of fine-tuning and distinguishing DyDiT from unrelated methodologies, such as sparse training. Additional experiments were provided, including comparisons to state-of-the-art models and evaluations under different conditions, to strengthen the claims.

While some concerns about incrementalism persist, the authors’ responses successfully clarified the scope and novelty of the work, and the demonstrated practical impact justifies its inclusion in the conference. The disagreements, particularly with TEwe, rooted more in methodological preferences rather than unresolved flaws in the submission. Considering the detailed rebuttals and the paper's contributions to improving diffusion model efficiency, the AC recommends acceptance.

**Additional Comments On Reviewer Discussion:**

Please refer to the meta review.

---

### Decision · Program_Chairs · 2025-01-22

Accept (Poster)